# From classical to quantum regime of topological surface states via defect engineering

Maryam Salehi, Xiong Yao and Seongshik Oh*

**Abstract:** Since the notion of topological insulator (TI) was envisioned in late 2000s, topology has become a new paradigm in condensed matter physics. Realization of topology as a generic property of materials has led to numerous predictions of topological effects. Although most of the classical topological effects, directly resulting from the presence of the spin-momentum-locked topological surface states (TSS), were experimentally confirmed soon after the theoretical prediction of TIs, many topological quantum effects remained elusive for a long while. It turns out that native defects, particularly interfacial defects, have been the main culprit behind this impasse. Even after quantum regime is achieved for the bulk states, TSS still tends to remain in the classical regime due to high density of interfacial defects, which frequently donate mobile carriers due to the very nature of the topologically-protected surface states. However, with several defect engineering schemes that suppress these effects, a series of topological quantum effects have emerged including quantum anomalous Hall effect, quantum Hall effect, quantized Faraday/Kerr rotations, topological quantum phase transitions, axion insulating state, zeroth-Landau level state, etc. Here, we review how these defect engineering schemes have allowed topological surface states to pull out of the murky classical regime and reveal their elusive quantum signatures, over the past decade.

# I.    Introduction

Abstract ideas in mathematics have often played a major role in our understanding of the physical world. Over the past decade and a half, the notion of topology was used to predict a new phase of matter, now known as a topological insulator (TI), with an insulating bulk and unusual metallic surfaces[1-7]. Unlike the related quantum Hall system, which requires broken time reversal symmetry (TRS), the band structures of TIs are protected by TRS. In three dimensions, both weak and strong topological insulators, differentiated by the number of Dirac cones (even or odd) on each surface, exist. In this review, we will discuss the latter, which we abbreviate simply as TI from here on. TIs possess electronic bulk bands with fundamentally different topology than their trivial counterparts, a characteristic that originates from spin-orbit-interaction-induced band inversion. The non-trivial topology of TIs is manifested by the existence of an odd number of gapless metallic surface states that exist at the interfaces between TIs and trivial insulators (including vacuum). These topological surface states (TSS) are protected against strong localization by TRS and host Dirac-like fermions whose spin and momentum are locked to each other. These features of TSS make TIs important for exploration of various new physics and applications.

In 2008, a topologically nontrivial phase was first discovered in the $Bi_{1-x}Sb_x$ alloy system using angle resolved photoemission spectroscopy (ARPES) and scanning tunneling spectroscopy (STS)[6,8-11]. This first-generation TI had a small bulk band gap and a complicated TSS band structure with five Dirac cones per surface. The focus of TI research has since shifted to chalcogenide compounds. The family of pnictogen chalcogenide compounds $Bi_2Se_3$, $Bi_2Te_3$, and $Sb_2Te_3$ were theoretically predicted[12,13] and subsequently observed by ARPES[14-18] and scanning tunneling spectroscopy/microscopy (STS/STM)[19-23] to be TIs. These compounds have a relatively large bulk band gap and a single TSS per surface, making them an excellent platform for the theoretical and experimental study of topological phases. An ideal TI has a

fully insulating bulk so that only the TSS partake in electrical conduction. It was soon realized, however, that these chalcogenide TIs have a highly conducting bulk due to crystal defects. The conducting bulk masks TSS conduction and impedes exploration of new physics and applications specifically attributed to TSS conduction. Therefore, a major focus in TI research has been to suppress this parasitic bulk conduction and realize TSS-dominated transport with proper defect engineering schemes.

Historically, defect engineering played key roles in bringing about major materials advancements, leading both to the observation of numerous novel phenomena and toward profound technological applications. Defect engineering consists of tailoring material's properties through different growth methodologies, such as impurity addition/alloying, intercalation, compensation doping, remote/modulation doping (to avoid impurity-induced scattering), and interface engineering. Success stories range from realizing exotic states of matter such as the fractional quantum Hall system to creating ubiquitous technologies such as complementary metal-oxide-semiconductor (CMOS) transistors. The fractional quantum Hall effect, awarded the 1998 Nobel prize, was first observed in a 2-dimensional electron gas (2DEG) formed with interface-engineered and modulation-doped GaAs/AlGaAs films[24-26]. The CMOS transistors, billions of which are crammed into modern computing chips, function based on gating a combination of n- and p-doped Si. The long sought-after blue light emitting diode (LED), awarded the 2014 Physics Nobel prize, required development of special interface engineering and doping schemes on GaN/AlGaN layers[27-29].

Likewise, similar defect engineering strategies were applied to TI systems to suppress the bulk and surface defects that have plagued both thin films and bulk crystals. Efforts include field effect modulation[30-39], growth of thin films with high surface to volume ratio on various substrates[30,40-43], compensation doping as well as intercalation[44,45], growth of ternary[46-53] and

quaternary[54-56] compounds derived from parent materials, and more recently, growth of TI thin films on an optimized buffer-layer through interface-engineering[57]. With these various defect-engineering techniques, the TI systems have evolved from bulk-dominant and conducting bulk[58], to TSS-dominant yet with conducting bulk[42], to TSS-only yet in the classical regime with insulating bulk[45], and finally to TSS-only in the quantum regime with insulating bulk[57,59,60]. With these developments, a plethora of novel phenomena in both classical and quantum regime of TSS have been discovered, including the quantum anomalous Hall effect[61-65] and its transition to an insulator[66,67], the quantum Hall effect[57,68,69,70] and its transition to a Hall insulator[60], the axion insulator[71-75], quantized Faraday and Kerr rotation[76,77], a finite-size driven topological and metal-insulator transitions[78], and the possible observation of chiral Majorana modes[79,80] as a potential platform for topological quantum computation[81].

Here, we review these developments in bulk crystals and thin films of chalcogenide TIs toward quantum regime of TSS, emphasizing the critical role of defect engineering in these efforts. In section II, we discuss the bulk conduction problem in $Bi_2Se_3$, $Bi_2Te_3$, and $Sb_2Te_3$, with particular focus on transport measurements of $Bi_2Se_3$ films on various substrates. Section III covers the efforts in materials engineering and success in suppressing bulk conduction through doping and growth of ternary and quaternary compounds. In particular, the critical role of interface engineering will be discussed on the way toward the extreme quantum regime of TSS. In section IV we will discuss observation of various quantum effects in TIs and magnetic TIs, made available via various defect engineering schemes. Finally, in section V we will conclude with outlook towards the future of TI research.

## II.    Bulk conduction problem in topological insulators

The heavy pnictide chalcogenides, materials in the form of $A_2X_3$, are layered compounds with a rhombohedral structure. The layers are arranged along the *c*-axis direction in the five atomic

layer-thick sequence X–A–X–A–X (A: Sb/Bi, X: Se/Te), which comprises quintuple layer (QL; 1 QL ≈ 1 nm; one unit-cell is composed of three QLs) where van der Waals forces bond adjoining QLs. These 3D TI materials, which are also known for being good thermo-electrics, possess simple TSS band structure with a single Dirac cone centered at $\Gamma$ point of the Brillouin zone. $Bi_2Se_3$ (lattice constant of 4.14Å) and $Sb_2Te_3$ (lattice constant of 4.25Å) have their Dirac point within the bulk bandgap, while the Dirac point in $Bi_2Te_3$ (lattice constant of 4.38Å) lies beneath the top of bulk valence band, which makes it intrinsically impossible to probe physics near the Dirac point in this material[15]. All these materials have sizeable bulk bandgap in the range of a few hundred meVs, making them suitable even for room-temperature applications[82,83].

It is known that these materials are (naturally) heavily doped due to crystal defects, such as vacancies and anti-site defects. $Bi_2Se_3$ and $Sb_2Te_3$ are naturally n- and p-type, respectively, while $Bi_2Te_3$ can be either p- or n-type depending on the growth conditions and defect types (vacancies or anti-sites). This native doping effect pushes the Fermi level ($E_F$) to bulk conduction (n-type) or valence (p-type) band, making their bulk states conducting. Furthermore, extra interfacial defects tend to form bulk-derived two dimensional electron gas (2DEG) near the surfaces[84-88]. These 2DEGs along with bulk conduction channels compete with the TSS and make it difficult to detect TSS via transport.

Bulk crystals of $Bi_2Te_3$, $Bi_2Se_3$, and $Sb_2Te_3$ are usually grown by modified Bridgeman technique[46,52,68,89,90]. In bulk $Bi_2Se_3$, for example, by varying Se:Bi ratio and temperature, crystals with wide range of bulk carrier density from $10^{16}$ to $10^{20}$ cm$^{-3}$ can be grown. However, even in the crystals with lowest bulk carrier concentration of $10^{16}$ cm$^{-3}$, TSS conduction could not be unambiguously detected from transport measurements[91]. In fact, analysis of Shubnikov de Haas oscillations in samples with carrier density as low as $10^{17}$ cm$^{-3}$ showed that these oscillations came from bulk rather than TSS[92]. Importantly, crystals from the same batch which

showed $E_F$ lying in the bulk band gap from ARPES study, still showed SdH oscillations from the bulk state. Such apparent discrepancy was explained as the manifestation of upward bending of the bulk bands near the surface, which results due to $E_F$ deep in the bulk lying in the bulk conduction band even though near the surface it lies in the bulk gap. The significant bulk conduction in even the lowest doped bulk crystals poses a considerable challenge in as-grown bulk crystals. This can be explained by calculating the Mott criterion[84,93]. In principle, the defects that are embedded in the crystal form atom-like bound states with an effective Bohr radius $a_B = \epsilon \frac{m}{m^*} a_0$, where $\epsilon$, $m$, $m^*$ and $a_0$ are the dielectric constant of crystal, free electron mass, effective mass of bulk electrons in the crystal, and free space Bohr radius ($\approx 0.05$ nm), respectively. For the case of Se vacancy in $Bi_2Se_3$, where $\epsilon \approx 110$ and $m^* \approx 0.15$ [92], $a_B$ is almost 37 nm. As the number of vacancies increases, they begin to overlap, and the electrons bound to vacancies become mobile. When this number reaches the critical dopant density ($N_C$), where the mean spacing between the dopants becomes of the same order as the effective Bohr radius, a metal-to-insulator transition occurs[94]. This critical value was quantified by Sir Neville Mott in 1960s as $N_C^{-\frac{1}{3}} \approx 4a_B$, which yields $N_C \approx 3 \times 10^{14}$ cm$^{-3}$ for $Bi_2Se_3$[84]. Based on this, it is not surprising that all bulk crystals have conducting bulk states; even the lowest ever reported volume carrier density of $\sim 10^{16}$ cm$^{-3}$ in $Bi_2Se_3$ bulk crystals[91], which is already two orders of magnitude higher than $N_C$, has a conducting bulk.

One natural solution for reducing the contribution from bulk is to thin the samples. In principle, this reduces the overall bulk carriers without affecting the TSS conduction except in ultrathin regime, where TSS becomes gapped due to hybridization[95]. Thin specimen from bulk crystals can be obtained by cleaving and transferring them to an arbitrary substrate for transport measurement. However, such processes tend to introduce additional defects, which significantly raise the carrier density of thin specimen while lowering the carrier mobility. An

alternative way of obtaining thin films is to grow them on a substrate. Although both chemical and physical deposition methods have been used in the past, the latter has been more frequently used due to their clean growth environment and simplicity. In physical deposition, sputtering, pulsed laser deposition (PLD) and molecular beam epitaxy (MBE) can be used, but so far MBE has produced the highest quality TI thin films.

MBE offers a wide gamut of advantages over other thin film growth techniques. First, the ultra-high vacuum environment during growth helps minimize foreign defects in the film. Second, MBE allows for precise thickness, doping control, and hetero-structure engineering over a large area. Third, *in situ* analysis tools such as reflection high energy electron diffraction (RHEED) allows for real-time monitoring of the film growth. Fourth, the self-limited growth mode, made possible by the volatility of the chalcogens, guarantees perfect stoichiometry for binary chalcogenides with ease. Finally, being layered materials, these chalcogenides can be grown on a wide variety of substrates due to van der Waals epitaxy. Due to these benefits, significant number of TI thin films have been grown by MBE technique on various substrates [96-110]. For instance, $Bi_2Se_3$, the most studied TI among the three chalcogenide compounds, has been grown on several substrates, such as $Al_2O_3$ (lattice mismatch of 15%)[42,111,112], Si (lattice mismatch of -7.3%)[40,58], GaAs (lattice mismatch of -3.4%)[113-116], InP (lattice mismatch of 0.2%)[117-119], CdS (lattice mismatch of -0.2%)[120], amorphous $SiO_2$ (complete lattice mismatch)[41,121,122], SiC (lattice mismatch of -26%)[123], graphene (lattice mismatch of -40.6%)[43,124,125], $SrTiO_3$ (lattice mismatch of 5.7%)[30,126], $MoS_2$ (lattice mismatch of 25%)[127], etc. However, all these films show much higher carrier-densities than are required for bulk insulating $Bi_2Se_3$ films. While various factors such as growth temperature, pressure, and cation to anion flux ratio play a role of varying degree in determining film quality, the choice of substrate itself seems to be the most important factor. For example, in $Bi_2Se_3$ films grown on $Al_2O_3$ (0001) subtrate, high and constant carrier density ($n_{sheet} \approx 3\text{-}4\times10^{13}$ cm$^{-2}$) is observed in

a wide range of film thickness, suggesting dominance of two-dimensional conduction channel[42]. Such thickness-independent sheet carrier density implies that interfacial defects dominate over any bulk defects on the chemically inert $Al_2O_3$ substrates. Bansal *et al.*[42] attributed this to combined effect of TSS and trivial 2DEG states. In contrast, carrier density in $Bi_2Se_3$ films grown on Si was found to scale with film thickness ($n_{sheet} \sim t^{0.5}$), suggesting significantly higher and varying bulk defect density in films grown in Si compared to those grown on $Al_2O_3$[58].

It is likely that such behavior stems from highly reactive nature of Si compared to $Al_2O_3$. In fact, the first MBE-growth of $Bi_2Se_3$ was on Bi terminated Si(111) substrate[128,129] (by depositing a monolayer of a $\beta$-phase Bi as buffer-layer for $Bi_2Se_3$ growth), which was inspired by an earlier work by Wan *et al.*[130] in 1991 where depositing a monolayer of Bi on Si (111) yielded either 1/3 ($\alpha$-phase) or a full monolayer ($\beta$-phase) coverage. An alternative way which gives a higher quality thin film with sharp interface is terminating the Si dangling bonds by exposing the Si substrate to Se flux above the Se-sticking temperature but substantially below the optimal growth temperature of $Bi_2Se_3$ to prevent chemical reaction between Si substrate and Se. In such a condition, only a monolayer of Se deposits on the Si surface and the excess Se desorbs without sticking (self-limited growth). $Bi_2Se_3$ film can then be grown on this Se treated surface in a two-step growth fashion where first, a thin seed-layer of $Bi_2Se_3$ (3QL) is grown at lower temperature. Then, by heating the sample to higher temperatures (~220˚C), the crystallinity of this thin seed-layer gets continuously better, serving as a template for the following $Bi_2Se_3$ layers[40]. Similar two-step growth scheme turns out to be also effective for other substrates as well[42].

In conventional systems like GaAs, lattice matching of the substrate is the most stringent constraint on the film growth due to strong bonding between the film and the substrate. However, for chalcogenide TIs with their layered nature, chemical reactivity turns out to be more important than the lattice matching for the film quality. For example, $Bi_2Se_3$ films grown

on amorphous $SiO_2$ substrate led to lower carrier density and higher mobility than the films grown on Si substrate, highlighting the importance of chemical compatibility over lattice matching[41]. Similarly, $Bi_2Se_3$ films grown on near perfect lattice matched InP or CdS (only with 0.2% and -0.2% lattice mismatch, respectively) substrates were even worse than those grown on poorly-lattice matched substrates such as $Al_2O_3$. The strong chemical reaction at the interface is likely the main cause for the degraded electrical properties in the $Bi_2Se_3$ films grown on these lattice-matched, yet chemically ill-matched substrates[119,120]. This conclusion is also supported by the observation that utilizing Se-sharing ZnCdSe buffer-layer on InP substrate led to much reduced sheet carrier density of $6-9\times10^{12}$ $cm^{-2}$ [131]. Furthermore, the significant drop in sheet carrier density after separation of $Bi_2Se_3$ film from $Al_2O_3$ substrate (using wet etching) and transferring it onto a Si or an STO substrate confirmed that the interface is the major source of defects[132]. Eventually, these observations led to the development of chemically- and structurally-matched $In_2Se_3$-based buffer-layers followed by extremely high-mobility ($> 15,000$ $cm^2V^{-1}s^{-1}$) and low-carrier density ($< 1\times10^{12}$ $cm^{-2}$) $Bi_2Se_3$ thin films, reported by Koirala, *et al* [57].

Lastly, in this section we discuss the relevant transport measurements in TI thin films. Significant amount of information about TIs can be extracted from low-temperature magneto-resistance measurements. The Hall effect along with magnetoresistance measurement provides carrier density, its type and mobility. Thickness-dependence of the carrier density tells about whether the channel is two- or three-dimensional in nature, as discussed above when comparing $Bi_2Se_3$ films grown on sapphire vs. silicon substrates. In samples with reasonably high mobility and small electron-hole puddle size, quantum oscillations show up. Such oscillations can give information about whether the oscillations arise from TSS, trivial 2DEG or bulk states. This is typically done by angle dependence of SdH oscillations and by deducing the phase of the oscillations. For TSS, which are Dirac-like electrons with π Berry phase, the oscillations result

in an additional phase of ½ (in units of 2π). However, the phase is frequently found to be around midway between 0 and ½), and this discrepancy has been normally attributed to Zeeman splitting resulting from high g-factor of electrons in $Bi_2Se_3$[133]. However, using SdH oscillations to identify the TSS has fundamental limitations. First of all, SdH oscillations cannot fully distinguish between trivial 2DEG states and TSS, because due to interfacial Rashba effect even trivial 2DEG state can still exhibit non-zero Berry phase in their SdH oscillations. Moreover, unlike Hall effect, which detects all mobile channels, SdH oscillations can be observed only from high and uniform mobility channels. Frequently, the carrier densities detected with SdH oscillations are much smaller than those measured by Hall effect. In fact, many of the claimed TSSs observed by SdH oscillations in the literature are more consistent with the trivial 2DEG states than TSS[42,134]. Only if the Fermi level is within the band gap both on the surface and in the bulk, and top and bottom surfaces have identically high mobilities, both SdH oscillations and Hall effect can probe TSS channels fully and consistently [45].

Another important transport property that can be used as a quantitative tool in TI thin films is weak anti- localization (WAL) effect. Weak anti-localization is the quantum correction to the classical conductance value in disordered systems with strong spin orbit coupling (SOC). The effect manifests itself as increase in conductance of the system with respect to its classical value due to destructive interference between time reversed partner electron waves, which lead to decreased probability for backscattering. In an applied magnetic field, the WAL effect diminishes quickly. In magnetoresistance measurements, this effect shows up as a cusp in resistance at zero field. While WAL is conventionally associated with strong SOC, which is present in TIs, a more powerful mechanism is responsible for the ubiquitous WAL observed in TIs. The Dirac surface states of TIs have a π Berry phase, so time-reversed paths always destructively interfere, and WAL is always observed. With the introduction of magnetic field, the phase coherence diminishes rapidly, and this shows up as a sharp increase in resistance

with magnetic field. In the two dimensional system (i.e. thickness $< l_\phi$), this qualitative description can be made quantitative by fitting the magneto-conductance data with Hikami-Larkin-Nagaoka (HLN) formula[135]: $\Delta G(B) = \frac{\tilde{A}e^2}{2\pi h}[ln\left(\frac{B_\phi}{B}\right) - \Psi\left(\frac{1}{2} + \frac{B_\phi}{B}\right)]$ , where $\tilde{A}$ corresponds to the number of independent conducting channels in the system (i.e. $\tilde{A}$ is 1 if there is a single channel). In TIs with conducting bulk, the usual scenario is that electrons/holes on the surface can couple to this bulk state, and therefore, the whole system acts like a single coherently coupled channel, resulting in $\tilde{A} = 1$[30,42,58,126,134,136] which is far from the ideal value $\tilde{A} = 2$ for true TI films with two, top and bottom, surface states. If, on the other hand, the bulk is insulating or if one of the TSS is decoupled from the bulk, then this results in $\tilde{A} = 2$. For more details regarding WAL theory in TIs, we refer to ref.[137]. In thin films whose bulk is insulating, yet whose TSS are coupled by tunneling through the bulk, the surface states again act as a single channel. In thin films, however, tunneling between the top and bottom surface states hybridizes their wave functions, opening a gap around the Dirac point and changing the Berry phase of the surface state bands. When the Fermi level is far from the gap, the system acts like a single WAL channel, resulting in $\tilde{A} = 1$. As the $E_F$ approaches the gap, a crossover of the quantum corrections to weak localization (WL) has been observed[138]. A crossover from WAL to WL also occurs when time reversal symmetry is broken through addition of magnetic dopant/layer[139,140].

Chen *et al.*[141] and Steinberg *et al.*[142] showed that upon depleting the bulk state near one surface using electrostatic gating, $\tilde{A}$ gradually increases from 1 to 2. Later on, Brahlek *et al.* succeeded in achieving $\tilde{A} = 2$ in bulk insulating Cu-doped $Bi_2Se_3$ films without relying on gating[45]. Brahlek *et al.* subsequently demonstrated that $\tilde{A} = 2$ can also be achieved by growing a tunable non-TI $(Bi_{1-x}In_x)_2Se_3$ layer between two TI $Bi_2Se_3$ layers[143]. In this heterostructure, the non-TI layer, when tuned to an insulating phase beyond a critical thickness, fully decouples the top and bottom TI $Bi_2Se_3$ layers (each acting as a single channel), resulting in total 2

conducting channels. More recently, Shibayev et al. have further demonstrated that $\tilde{A}$ scales nearly linearly with the number of interface pairs in MBE-grown superlattices of the TI $Bi_2Se_3$/normal insulator $In_2Se_3$[144].

## III.   Defect engineering schemes for topological insulators

When it comes to TIs, there are two types of defects that need to be considered: bulk defects and surface defects. In principle, the Mott criterion gives critical information about the position of the Fermi level ($E_F$) deep inside the bulk. As we mentioned above, all the experimental transport data give bulk carrier densities that exceed the Mott criterion by at least a few orders of magnitude, which then implies that the $E_F$, deep in the bulk, must be at the conduction band minimum (or valence band maximum for p-type).  Specifically, $N_C \approx 10^{14}$ cm$^{-3}$ (calculated in the previous section) for TIs is orders of magnitude smaller than, for instance, $N_C \approx 10^{18}$ cm$^{-3}$ for Si and GaAs, suggesting that it may be thermodynamically impossible to reach such low defect density in bulk crystals of chalcogenide TI materials because of their much weaker chemical bonding. Now, the important question is whether it is possible at all to bring the Fermi level in the bulk gap and achieve a true bulk-insulating TI.

To answer this question, it is important to understand the physics of surface and in particular the band bending near the surface. Based on the Mott criterion, let us assume that the bulk $E_F$ is pinned to the bottom of the conduction band (CB). Now, if the surface $E_F$ is different than the bulk $E_F$, then the charges keep flowing until the Fermi level is aligned everywhere in the material, which results in band-bending near the surface and thus a spatial charge imbalance near the surface.  The Fermi wave vector $k_F$ for a 2-dimensional system is given by $k_F = (2\pi n_{2D})^{\frac{1}{2}}$, where $n_{2D}$, the 2-dimensional carrier density for two topological surface states of a 3D TI, is expected to be $n_{2D} = 2 \times n_{SS}$ (if we assume that each surface has identical carrier density of $n_{SS}$). Then, based on the dispersion relation for a Dirac state, the energy of TSS as

measured from the Dirac point is $E_{SS} = \hbar v_F (4\pi n_{SS})^{\frac{1}{2}}$, where $v_F$ is the Fermi velocity. If we

calculate $n_{ss}$ for Bi$_2$Se$_3$ with its conduction band minimum at almost 200 meV above the Dirac

point and $v_F \approx 4 \times 10^5$ ms$^{-1}$ (extracted from the ARPES data), then for the surface $E_F$ to be

located at the CB minimum to fulfill the flat band scenario, $n_{SS}$ needs to be $\sim 5 \times 10^{12}$ cm$^{-2}$ (or

$n_{2D} \approx 1 \times 10^{13}$ cm$^{-2}$).

However, if $n_{ss} > 5 \times 10^{12}$ cm$^{-2}$ (or $n_{2D} > 1 \times 10^{13}$ cm$^{-2}$), then electrons flow from

the surface towards the nearby bulk until $E_F$ is aligned everywhere. This gives the surface a net

positive charge and the nearby bulk a net negative charge, and thus causes the bands bend

downward near the surface, creating an accumulation region near the surface. The downward

band-bending is typically what happens for as-grown TI samples, especially once the sample

is exposed to air, and gives rise to a non-topological 2DEG which is also observable in ARPES.

If, on the other hand, $n_{ss} < 5 \times 10^{12}$ cm$^{-2}$, then electrons flow from the nearby bulk to the

surface which results in upward band-bending, and thus a depletion region near the surface.

The accumulation region caused by the downward band-bending can be treated like a typical

metal, and thus Thomas-Fermi approximation can be used to estimate the screening length

scale $l_s \approx \left( \frac{\varepsilon_0 \pi^2 \hbar^2}{k_F m^* e^2} \right)^{\frac{1}{2}}$ to be less than 1 nm, where $e$ is the electron charge, $\varepsilon_0$ is the free space

dielectric constant and the Fermi wave vector $k_F \approx 0.07$ /Å as well as the effective mass $m^* \approx$

$0.15 m_e$ are extracted from ARPES measurements. However, the story is completely different

for the depletion region in the upward band-bending scenario, where Poisson equation should

be used to find the depletion screening length, which is usually much larger than the length for

downward band-banding.

Although, downward band-bending is what often occurs in TIs, Brahlek et al.[84] have

shown that if upward band-bending - which is not easy but possible - is achieved for TI thin

films, then the Mott criterion, which is responsible for the conducting bulk, can be

circumvented once the films are made thinner than approximately twice the depletion region. This can qualitatively be explained by the fact that the upward band-bending is the result of charge transfer from the nearby bulk to the surface: if the film thickness is thinner than depletion region, then there are not enough mobile charges left in the nearby bulk to be transferred to the surface to equilibrate the $E_F$. Thus, the only way for the system to do so is to allow the bulk $E_F$ to fall below the CB minimum, and hence true bulk-insulating TIs can be achieved[45,57,60]. Using Poisson equation $\nabla^2 V(z) = -\frac{e^2 N_B}{\varepsilon\varepsilon_0}$, where $V$ is the potential energy as a function of distance from the surface ($z$), $e$ is the electron charge, and $N_B$ is the bulk dopant density (assumed to be uniformly distributed), it can be shown that $\Delta V = e^2 z_d^2 N_B/(2\varepsilon\varepsilon_0)$ for the boundary conditions of $V(z = z_d) = 0$, and $V(z = 0) = \Delta V$, where $\Delta V$ is the energy difference between the bands deep in the bulk and at the surface, and $z_d$ is the depletion region length (no electric field beyond $z = z_d$). Then, by using reasonable parameters[42,45] in Bi2Se3, Brahlek *et al.* have estimated the depletion region to be ~50-100 nm thick. Films thinner than this length scale still preserve their topological nature, as long as the thickness remains above a critical thickness where the top and bottom surfaces start to hybridize[95,145-147], and thus a true TI with insulating bulk and $E_F$ in the bulk gap can be achieved in these thin films with upward bend-bending near the surfaces. Note that the bulk cannot be made insulating by thinning the films if there are accumulation layers on the surfaces due to downward band-bending.

Accordingly, in order to achieve bulk insulating TI thin films, it is essential to achieve surface depletion layers, and the first step toward that is to suppress bulk and interfacial defects, which is the subject of the following subsections. To suppress defects in a TI system, variety of techniques have been implemented in growth of both thin films and bulk crystals. Some of the most important growth methodologies for obtaining defect-suppressed TIs will be discussed below.

III.a    Compensation doping

One of the ways to eliminate defects is through compensation-doping[56,148-151]. In bulk $Bi_2Se_3$ crystals, compensation doping with group II elements, such as Ca has been used to lower the carrier density and to even tune the sample from n- to p-type[56,150,151]. However, such doping has been difficult to implement in MBE-grown $Bi_2Se_3$ thin films. Only recently with the use of special interfacial layers and proper capping layer, a systematic and reliable carrier tuning has been achieved[59]. This will be discussed in detail in later sections.

Interestingly, Cu doping was found to work as compensation dopants for $Bi_2Se_3$ thin films[45,152] (grown on sapphire substrate). Cu doping had previously been shown to induce superconductivity in bulk $Bi_2Se_3$, where the increase of carrier density upon Cu doping at low temperature was explained as one of the possible reasons for superconducting transition at 3 K [153,154]. However, in $Bi_2Se_3$ thin films, Cu doping functioned more like p-type and around an optimal doping of ~2% lowered the sheet carrier density from $3 \times 10^{13} cm^{-2}$ to $~5 \times 10^{12} cm^{-2}$, indicating much lower $E_F$ compared to pure $Bi_2Se_3$. Subsequently, WAL effect, for the first time, resulted in two conducting channels in these films (when thicker than ~10-20 QL)[45]. Furthermore, for the first time in TI films, SdH oscillation accounted for the entire carrier density measured by Hall effect, and together with WAL, ARPES[45] and terahertz measurements[152] confirmed that a true bulk-insulating TI with decoupled TSS is realized. In these films, majority of Cu dopants were found to be electrically neutral, so the decrease in carrier density is likely due to Cu dopants reducing the Se vacancies and/or somehow alleviating the interfacial defects rather than some compensation effect.

III.b    Ternary and quaternary compounds and isovalent alloying

Another way to suppress defects in topological materials is to grow ternary ($A_xB_{2-x}C_3$ or $A_2C_{3-x}D_x$, where A and B are pnictogens and C and D are chalcogens)[47,52,54] or quaternary compounds

$(A_{2-x}B_xC_yD_{3-y})$[54,56,155,156]. Such isovalent alloys, derived from $Bi_2Se_3$, $Sb_2Te_3$ and $Bi_2Te_3$, have resulted in major advances in the quality of topological materials, particularly in bulk crystals. As $Sb_2Se_3$ is topologically trivial, care must be taken so that the alloy composition maintains the topological nature of the material and the surface states remain intact. These compounds are useful in tuning the defect chemistry of the material, which can lead to decrease in carrier density and suppressed bulk conduction. Additionally, application of electrostatic gating helps with further depletion of carriers and lowering $E_F$ towards the Dirac point. Early on, by alloying $Bi_2Se_3$ with Sb, bulk crystals with low volume carrier density of $n_{3d} \approx 2$ to $3 \times 10^{16}$ cm$^{-3}$ were obtained leading to detection of quantum oscillations originating from surface states[157]. Although Sb does not directly deplete the n-type carriers due to its iso-valency with Bi, it helps lower the carrier density by reducing the Se vacancies.

Additionally, some of these compounds have led to bulk-insulating states. For example, in $(Bi_{1-x}Sb_x)_2Te_3$[47], substituting Sb for Bi pulls the Dirac point of $Bi_2Te_3$ from beneath the bulk valence band into the bulk band gap. Because $Bi_2Te_3$ (in this case) is n-type and $Sb_2Te_3$ is intrinsically p-type, $E_F$ in the solid solution can be tuned into the bulk band gap for the optimal range of $0.75 < x < 0.96$ with the lowest sheet carrier density of $n_{2d} \approx 1 \times 10^{12}$ cm$^{-2}$ achieved for $x = 0.96$[47]. Similar results are also reported by other groups [133,158-161] and in bulk crystals [32].

III.c    Interface-engineered TI films with record-low carrier density and record-high mobility

Interface-engineering is another effective approach to eliminate defects, which resulted in TI thin films with record-low carrier density and record-high carrier mobility thin film[57], and eventually enabled observation of various quantum effects of TSS, as discussed in the next section.

As it was discussed in the previous section, TI thin films can be grown on a wide range of substrates. However, all the commercially available substrates lead to high density of

interfacial defects. It turns out that structurally and chemically compatible insulating buffer layers are needed to suppress these defects. In this regard, Koirala et al. [57] reported the growth of record-low-carrier density and record-high-mobility $Bi_2Se_3$ thin films using $In_2Se_3$/$BiInSe_3$ (BIS) buffer layers. Both $In_2Se_3$ and $BiInSe_3$ are insulating, chemically inert, and share similar structure with $Bi_2Se_3$ with 3.3% and 1.6% lattice mismatch, respectively: although there are much-better lattice-matched commercial substrates such as InP and CdS, the chemical and structural matching provided by the BIS buffer layers turns out to be much more important in suppressing the interfacial defects. In parallel, solid solution $(Bi_{1-x}In_x)_2Se_3$ and its topological phase transition from TI to a trivial insulator (as a result of weakening SOC-strength upon adding a lighter element In)[39,78,162-165] as well as artificial topological phases composed of $Bi_2Se_3$/$In_2Se_3$ superlattices[143,144] have also been extensively studied.

The growth of high-quality BIS buffer layer requires a non-standard growth scheme (see Fig. 1). First, a very thin layer (~3 QL) of $Bi_2Se_3$ is grown on an $Al_2O_3$ (0001) substrate as a template for the $In_2Se_3$ layer, because $In_2Se_3$ does not grow well on sapphire due to the presence of multiple phases. Then, upon heating, the thin $Bi_2Se_3$ layer evaporates and diffuses out of the $In_2Se_3$ layer, leaving behind an insulating $In_2Se_3$ layer for the following growth of $BiInSe_3$ layer. Unlike growths on commercial substrates, high-angle annular dark-field scanning transmission electron microscopy (HAADF-STEM) shows atomically sharp interfaces between the film and the BIS buffer layer, which is indicative of suppressed interfacial defects (Fig. 1b-e).

Suppression of interfacial defects in TI films can be best probed in transport measurement due to its sensitivity to the concentration of defects. The Hall resistance data, including the sheet carrier density ($n_{sheet}$) and carrier mobility ($\mu$), of $Bi_2Se_3$ films grown on BIS buffer layer are compared with the films grown on $Al_2O_3$ (0001) and Si (111) in Fig. 1f and g. In Fig. 1f, $n_{sheet} \approx 1 - 3 \times 10^{12}$ cm$^{-2}$ of $Bi_2Se_3$ films on the BIS buffer layer is about an

order of magnitude smaller than those on $Al_2O_3(0001)$ or Si (111) substrates. The mobility of buffer-layer-based $Bi_2Se_3$ film is also about an order of magnitude higher than any previously obtained values, reaching as high as $\mu \approx 16,000$ cm$^2$/Vs for a 50 QL-thick film (Fig.1g).

No less important is the capping layer, because exposure to air can significantly change the carrier density of TI films over time: this is particularly more so for low carrier density TI films[166]. Also for surface-sensitive techniques, such as ARPES or STM, which require pristine film surface[167], a capping layer that can be removed in a vacuum chamber through heating (like Se or Te capping) is a must. Additionally, in the case of $Bi_2Se_3$, a charge-depleting capping layer, such as molybdenum trioxide ($MoO_3$ with high electron affinity and a large band-gap of ~3eV) can be exploited to not only protect the film against environmental degradation but also further lower $E_F$, which can be thought of as a natural gating[57,168]. Although a capping layer induces scattering and lowers the mobility, the mobility remains in acceptable range to observe most of the TSS-related phenomena. The lowest $n_{sheet}$ that was achieved in $MoO_3$-capped (with an extra Se-capping for further protection) $Bi_2Se_3$ films on the BIS buffer layer is $7 \times 10^{11}$ cm$^-$$^2$ in which TSS-originated QHE was observed and will be explained in more detail in the next section.

Applying a similar buffer-layer scheme (Fig. 1j-k) along with compensation doping to $Sb_2Te_3$ has led to the lowest sheet-carrier density of all time in any TI system, which was essential to reveal extreme quantum signatures of TSS as discussed in the next section. Since an early scanning tunneling spectroscopy (STS) study, it has been known that Dirac point of $Sb_2Te_3$ is better separated from the bulk bands compared to other TI materials such as $Bi_2Se_3$ and $Bi_2Te_3$. However, this fundamentally superior band structure of the $Sb_2Te_3$ system has not been properly utilized in transport studies due to its strong intrinsic nature of p-type doping. Salehi *et al* found a solution to this problem by growing titanium-doped $Sb_2Te_3$ thin films on $In_2Se_3/(Sb_{0.65}In_{0.35})_2Te_3$ buffer-layers, capped by $(Sb_{0.65}In_{0.35})_2Te_3$ layers: these films led to n-p

tunable ultra-low carrier density TI films (as low as $1.0 \times 10^{11}$ cm$^{-2}$). $(Sb_{1-x}In_x)_2Te_3$ is a solid solution of trivial insulator $In_2Te_3$ and topological insulator $Sb_2Te_3$. However, since $In_2Te_3$ has a different structure (defective zinc blende lattice with a = 6.15$\overset{\circ}{A}$) than $Sb_2Te_3$, $Sb_2Te_3$ structure cannot be maintained beyond a certain In concentration[169]. Therefore, optimized In concentration in the buffer and capping $(Sb_{1-x}In_x)_2Te_3$ layer is determined such that it is fully insulating while maintaining the $Sb_2Te_3$ structure. Using these interface-engineered $Sb_2Te_3$ films with ultra-low-carrier density, it has become possible to reach the extreme quantum signatures of TSS, which will be discussed in the next section.

Finally, the low-temperature transport properties (extracted from DC transport) of $Bi_2Se_3$, $Bi_2Te_3$, $Sb_2Te_3$, and some of their compounds in the form of thin films are summarized in Table I.

**Table I: DC transport properties of TI films**

| TI | Substrate | Thickness (nm) | $n_{sheet}$ ($10^{12}$cm$^{-2}$) | $\mu$ (cm$^2$V$^{-1}$s$^{-1}$) | Ref. | comment |
|---|---|---|---|---|---|---|
| $Bi_2Se_3$ | Si(111) | | -0.2 | 750 | 170 | Sb doped Nanoribbons with gating Zinc oxide capping |
| $Bi_2Se_3$ | Si(111) | 10 | -15 | 200 | 58 | |
| $Bi_2Se_3$ | Si(111) | 200 | -60 | 2000 | 128 | |
| $Bi_2Se_3$ | $\alpha$-SiO$_2$ | 20 | -34 to -48 | - | 122 | Back gating -50 V $\leq$ V$_G$ $\leq$ 50 V |
| $Bi_2Se_3$ | $\alpha$-SiO$_2$ | 7 | -22 | | 107 | |
| $Bi_2Se_3$ | GaAs(111) | 20 | -40 | 520 | 131 | |
| $Bi_2Se_3$ | CdS(0001) | 10 | -13 (impurity) -0.4 (TSS) | 380 (impurity) 5000 (TSS) | 158 | |
| $Bi_2Se_3$ | Al$_2$O$_3$(0001) | 20 | -62 | 807 | 171 | Se capping |
| $Bi_2Se_3$ | Al$_2$O$_3$(0001) | 10 | -2.6 (TSS) -38 (Bulk) | 1600 (TSS) 540 (Bulk) | 134 | |
| $Bi_2Se_3$ | Al$_2$O$_3$(0001) | 8-256 | -8 (2DEG) -30 (Bulk) | 3000 (2DEG) 500 (Bulk) | 42 | |
| $Bi_2Se_3$ | Al$_2$O$_3$(0001) | 20 | -5 | 2000 | 45 | Cu doping |
| $Bi_2Se_3$ | Al$_2$O$_3$(0001) | 15 | -28.5 | 426 | 96 | |
| $Bi_2Se_3$ | Al$_2$O$_3$(0001) | 20 | -70.6 | 650 | 172 | |
| $Bi_2Se_3$ | Al$_2$O$_3$(0001) | 5 | -18.9 | 316.6 | 173 | Se capping |
| $Bi_2Se_3$ | SrTiO$_3$(111) | 10 | +3.2 to -32 | 1000 | 30 | Back gating -150 V $\leq$ V$_G$ $\leq$ 50 V |
| $Bi_2Se_3$ | h-BN(0001) | 10 | -5.4 to -8.5(2DEG) & +0.011 to -8.3 (TSS$_{bottom}$) | | 174 | |
| $Bi_2Se_3$ | MgO(100) | 15 | -26.3 | 334 | 96 | |
| $Bi_2Se_3$ | Cr$_2$O$_3$(0001) | 15 | -27.6 | 108 | 96 | |

| | | | | | | |
|---|---|---|---|---|---|---|
| $Bi_2Se_3$ | $In_2Se_3$/BiInSe$_3$ | 8 | -0.7 | 4000 | 57 | MoO$_3$+Se capping, QHE |
| $Bi_2Se_3$ | $In_2Se_3$/BiInSe$_3$ | 15 | -1 | 16000 | 57 | No-capping; Measured immediately after growth |
| $Bi_2Te_3$ | $Al_2O_3$(0001) | 20 | -83.4 | 912 | 172 | |
| $Bi_2Te_3$ | Si(111) | 4 | -120 | 35 | 175 | |
| $Bi_2Te_3$ | Si(111) | 6 | -110 to -330 | - | 176 | |
| $Bi_2Te_3$ | SrTiO$_3$(111) | 15 | -3.8 | 1600 | 177 | 6 nm $Al_2O_3$ capping |
| $Bi_2Te_3$ | $Al_2O_3$(0001) | 15 | -9.5 | 1206 | 177 | 6 nm $Al_2O_3$ capping |
| $Sb_2Te_3$ | $Al_2O_3$(0001) | 5 | +7 | 200 | 47 | Te cap |
| $Sb_2Te_3$ | Si(111) | 9.6 to 45 | +44 to +67 | - | 176 | $Sb_2Te_3$/ $Bi_2Te_3$ heterostructures also studied |
| $Sb_2Te_3$ | Si(111)/SiO$_2$ | 21 | +111.3 | 4000 | 178 | |
| $Sb_2Te_3$ | Si(111) | 13.6 | +50 | 1000 | 179 | Gating $V_G$ = 10 V |
| Ti doped $Sb_2Te_3$ | $In_2Se_3$/ (Sb$_{0.65}$In$_{0.35}$)$_2$Se$_3$ | 8 | +0.1 | | 60 | QHE |
| Ti doped $Sb_2Te_3$ | $In_2Se_3$/ (Sb$_{0.65}$In$_{0.35}$)$_2$Se$_3$ | 8 | -0.14 | | 60 | QHE |
| (Bi$_x$ Sb$_{1-x}$)$_2$Te$_3$ | GaAs(111)B | | -0.7 | - | 180 | With $V_G$ = 2.5 V and for x = 0.53 |
| (Bi$_{1-x}$Sb$_x$)$_2$Te$_3$ | $Al_2O_3$(0001) | 5 | +1 | 550 | 47 | Te capping, x = 0.96, n-p tunability |
| (Bi$_{1-x}$Sb$_x$)$_2$Te$_3$ | SrTiO$_3$(111) | 20 | 3 (ungated) | 100 to 500 | 158 | Gated samples, n-p tunability, minimum carrier density was found for x = 0.5 and n to p transition for x ~ 0.35–0.45. |
| (Bi$_{1-x}$Sb$_x$)$_2$Te$_3$ | InP(111) | 20 | | | 160 | Ionic-liquid gate tunability from n- to p-type; the lowest residual charge carrier density at x ~ 0.8–0.9. |
| (Bi$_{1-x}$Sb$_x$)$_2$Te$_3$ | Si(111) | | +5 | 150 | 159 | Gate tunable n- to p-type transition. Lowest $n_{2D}$ ($5 \times 10^{12}$ cm$^{-2}$) and highest sheet resistance for $x = 42\%$. |

# IV.    Quantized signatures of topological surface states

IV.a    TSS-originated quantum Hall effect and the zeroth Landau level

The first quantum Hall effect (QHE) from TSS was observed in gated BSTS crystal flakes (Fig. 2a-b)) in 2014 [68], and a year later QHE was also observed in gated (Bi$_{1-x}$Sb$_x$)$_2$Te$_3$ (BST) thin films[69] (Fig. 3a-d).  In 2016, Xu *et al.* utilized dual (top and bottom) gating on BSTS crystals and performed more in-depth studies on the QHE of TSS[70] (Fig. 2c-f). In these studies they observed a series of ambipolar two-component half-integer Dirac quantum Hall states and signatures of the zeroth Landau level (LL) physics[181]. Nonetheless, many of the predicted

features of the zeroth LL such as topological magneto-electric effects[182,183] or excitonic superfluidity[184] still remain elusive.

The first QHE in a binary TI was reported in 2015 on interface-engineered $Bi_2Se_3$ films (Fig. 3e-h). Koirala *et al.* grew $Bi_2Se_3$ films without any impurity addition on BIS buffer layers and capped them with $MoO_3$/Se layers, and these have led to a very low carrier density of $n_{sheet}$ $\approx 7.0 \times 10^{11}$ cm$^{-2}$ and high mobility of $\mu \approx 4000$ cm$^2$V$^{-1}$s$^{-1}$. On these films, even without any gating, they observed perfect QHE with $R_{Hall} = 1.00000 \pm 0.00004$ ($\frac{h}{e^2}$) and vanishing longitudinal resistance ($R_{sheet} \approx 0.0 \pm 0.5$ Ω) when the applied magnetic field exceeds 25 T. The signature of QHE persisted even up to 70 K[57].

Further advance in QHE was achieved with interface engineered $Sb_2Te_3$ thin films in 2019 (Fig. 4). As mentioned in the previous section, with the use of compatible buffer and capping layers along with Ti counter-doping, Salehi *et al.* achieved an ultra-low carrier density of $1.0 \times 10^{11}$ cm$^{-2}$ in $Sb_2Te_3$ thin films[60]. On these films, they achieved QHE at fields as low as ~5 T without any gating (Fig. 4b). Furthermore, in contrast with the $Bi_2Se_3$ system, in which QHE did now show up in the p-type films[20,23,59], QHE showed up in both n- and p-type $Sb_2Te_3$ films, confirming the earlier Landau level spectroscopy studies[20,145]. The transport properties of these ultra-low-carrier density $Sb_2Te_3$ films remained almost the same over a year after the growth, suggesting the robustness of this platform for long term applications.

Furthermore, this ultra-low-carrier density platform allowed much cleaner access to zeroth LL than before. In particular, a magnetic field-driven quantum phase transition from QH to insulator phase with gigantic magnetoresistance ratio (as large as $8 \times 10^6$ % under 45 T) was observed[60] (Fig. 4c). Although QH-to-insulator transition (QIT) is well studied in conventional 2DEGs, this is the first observation of a magnetic-field-driven QIT in a TI system. Interestingly, scaling analysis (Fig. 4d) of this QIT transition revealed that it belongs to a different

universality class than that of the conventional 2DEGs, by providing the first case of dynamical critical exponent being two instead of one.

## IV.b    Magnetic topological insulators and quantum anomalous Hall effect

The gapless topological surface states of TIs are protected by TRS. On the other hand, a gap can open at the Dirac point, if TRS is broken either by proximitized magnetism (by growing a TI on a magnetic layer or vice versa) or by intrinsic magnetism, say, by doping TI films with magnetic ions [185-188]. The strong SOC of TIs, which leads to band inversion and thus the emergence of TSS, combined with the magnetic gap at the Dirac point could give rise to the quantum anomalous Hall effect (QAHE)[189-192].

Like the $v = 1$ QH system, the QAH system features an insulating two-dimensional bulk and a single chiral edge mode that conducts along the one-dimensional boundary of the system, resulting in quantized longitudinal and Hall conductances: $\sigma_{xx} = 0$ and $\sigma_{xy} = \pm\frac{e^2}{h}$ (or, equivalently, $\rho_{xx} = 0$ and $\rho_{xy} = \pm\frac{h}{e^2}$). Unlike QHE, QAHE requires no external magnetic field. Inspired by Duncan Haldane's 1988 proposal for QHE without Landau levels (LL)[193], which shared the 2016 Nobel prize, QAHE was predicted in 2008 in doped 2D TI HgTe quantum wells[185]. It was soon found, however, that magnetically doped HgTe samples exhibit paramagnetism, rather than ferromagnetism[194].

It was subsequently predicted that ferromagnetic (FM) order can be induced by the van Vleck mechanism in thin films of pnictogen chalcogenide TIs when doped with a proper transition metal element, either Cr or Fe[186]. They predicted that, in thin magnetic TI films (having hybridized surface states), QAHE could be realized when the FM exchange energy exceeds the hybridization gap. Initially, $Bi_2Se_3$ was predicted to be a promising platform to realize the QAHE, as the Dirac point of its surface state is within its large bulk band gap of 0.3

eV. QAHE, however, was never realized in magnetically-doped $Bi_2Se_3$ thin films, despite much effort. Only a small anomalous Hall effect (AHE) of a few Ohms was seen in V-doped $Bi_2Se_3$ samples[195] and, more recently, in low-carrier density buffer-layer-based $Bi_2Se_3$ thin films with Cr modulation doping[196]. The lack of QAHE in $Bi_2Se_3$ may be due to relatively weak SOC strength of the Se atoms: upon magnetic impurity substitution into the Bi sites, SOC weakens and the compound becomes a trivial insulator[197,198]. In contrast, Te-based TIs remain more robust against such substitution because of the stronger SOC strength of the Te atoms. Supporting this interpretation, an experiment by Zhang *et al.*[199] showed that AHE in $Cr_{0.22}Bi_{1.78}(Se_xTe_{1-x})_3$ thin films becomes weaker with increasing Se content, disappearing at x = 0.67. Furthermore, at x = 0.67, electrostatic gating restored the AH loop through the Stark effect. Here, when the material is already close to the topological-to-trivial quantum phase transition, a perpendicular electric field can shift the bands sufficiently to drive the system across the critical point. A ferromagnetic-to-paramagnetic phase transition coincides with the topological-to-trivial phase transition because the van Vleck mechanism is stronger in the topological phase [197,200-202]. Since topological non-triviality is more robust to impurity doping in Te-based materials than in Se-based materials[198,203], the search for a quantum anomalous Hall insulator was focused more on Te-based topological insulators.

QAHE was finally realized in 2013 for the first time [61], 132 years after the discovery of AH effect[204] (Fig. 5a-c). Using the ternary compound $Bi_xSb_{2-x}Te_3$ magnetically doped with Cr, $E_F$ was brought near the Dirac point by adjusting the Bi/Sb ratio. The films were grown by MBE on a $SrTiO_3$ dielectric substrate, and the chemical potential was finely tuned into the magnetic exchange gap by electrostatic gating. At 25 mK, a quantized Hall resistance ($\frac{h}{e^2} \approx$ 25.8k$\Omega$), concurrent with small longitudinal resistance ($0.098\frac{h}{e^2}$), was observed at zero external magnetic field. Application of a 10 T magnetic field reduced the longitudinal resistance to the

noise level. Additionally, using X-ray studies, Ye *et al.*[205] showed that the interaction between Sb/Te *p* and Cr *d* orbitals is crucial to the long-range magnetic order in these films.

The 2D TSS bands are gapped by the magnetic exchange interaction. One would expect the exchange bandgap to be comparable to the Curie temperature of the QAH insulator, which is typically tens of Kelvin. Yet, the Arrhenius scaling of thermally activated dissipation indicates that the bandgap is only around 1 K[206]. This discrepancy is explained by smearing of the bandgap by disorder[66]: the effective bandgap of the material is reduced from the clean-limit value by the spatial fluctuations of the band edge. Potential sources of such a disorder include crystalline defects, magnetic dopant clustering, and surface nonuniformity. In addition, dissipation due to superfluous conduction along device edges have been predicted[207]. As a potential source of dissipation, Cr-dopant clustering has attracted much attention following STM imaging revealing inhomogeneity in the atomic positioning of Cr dopants[208]. Areas with higher concentrations of Cr dopants were found to have a proportionally higher Dirac mass, indicating that ferromagnetism is a local effect in Cr-doped TIs. A nanoscale magnetic imaging study found that the magnetism of these materials consists of small (order of tens of nanometers) weakly interacting superparamagnetic domains[209]. These magnetic islands perhaps correspond to clusters of Cr dopants. Finding new ways of magnetic doping is therefore an important aspect for realizing a QAH system at higher temperatures.

In 2015, a modulation doping scheme, where the magnetic dopants are concentrated in 1 nm-thick Cr-rich layers, has also allowed the QAHE to be seen at higher temperatures. Mogi *et al*. grew a penta-layer heterostructure of (1nm $(Bi_{0.22}Sb_{0.78})_2Te_3$/1nm $Cr_{0.46}(Bi_{0.22}Sb_{0.78})_{1.54}Te_3$/4nm $(Bi_{0.22}Sb_{0.78})_2Te_3$/1nm $Cr_{0.46}(Bi_{0.22}Sb_{0.78})_{1.54}Te_3$/1nm $(Bi_{0.22}Sb_{0.78})_2Te_3$) on a InP(111) substrate, observing QAHE at temperatures up to almost 2 K (the signature of QAH remains up to 4.2k)[63]: Fig. 5g-j. Shortly after, QAHE was also realized in V-doped $Bi_xSb_{2-x}Te_3$ films[51] (Fig. 5d-f). V-doped films were not initially considered a

candidate QAH insulator because simulations predicted the formation, upon introduction of substitutional V dopants, of *d*-orbital impurity bands at the Fermi energy[186]. Nevertheless, a $V_{0.11}(Bi_{0.29}Sb_{0.71})_{1.89}Te_3$ film exhibited QAHE with Hall conductivity $(0.9998 \pm 0.0006) \frac{e^2}{h}$ and nearly vanishing longitudinal resistivity $(3.35 \pm 1.76)\ \Omega$ at 25 mK and zero field[181]. Later on, using metrological equipment, the Hall resistance in a 9 nm-thick film of $V_{0.1}(Bi_{0.21}Sb_{0.79})_{1.9}Te_3$ was quantized to $\frac{h}{e^2}$ within an uncertainty of about a half part-per-million[210]. V-doped BST films have a higher coercive field than Cr-doped films (~1 T versus ~150 mT at dilution refrigeration temperatures), indicating larger perpendicular magnetic anisotropy, and may be more uniformly doped, providing more homogeneous magnetism and Dirac mass[211].

In addition, co-doping has been proposed as a step towards higher temperature QAH systems[212]. Before this, it was predicted that proper co-doping could enhance magnetism in diluted magnetic semiconductors[213]. Using Cr and V co-doping, a 5 QL-thick MBE-grown $(Cr_{0.16}V_{0.84})_{019}(Bi_xSb_{1-x})_{1.81}Te_3$ film demonstrated $\rho_{xy} = \frac{h}{e^2}$ and $\rho_{xx} = 0.009 \frac{h}{e^2}$ at 300 mK[211] where $T_c = 25$ K. In fact, Cr and V co-doping was previously implemented in $Sb_2Te_3$ bulk crystals in 2007. Drašar *et al.* observed a significant enhancement in remnant magnetization when Cr is added to V-doped $Sb_2Te_3$ where $T_c$ remained comparable to that of only V-doped $Sb_2Te_3$. In contrast, modulation doped films with a Cr-rich layer and a V-rich layer reach an altogether different state. Because Cr- and V-doped TIs have different coercivities, the magnetization of the Cr-rich layer and the V-rich layer flip at different external fields. Between the two coercive fields, the magnetization of the two layers points oppositely[71,73]. The resulting state, known as an axion insulator, features gapped surfaces without edge states. In the axion insulator state, the surfaces of the TI are predicted to exhibit a topological magnetoelectric effect, wherein an applied magnetic field produces an electrical polarization and vice versa[182,183].

The extremely low temperature ($\leq$ 2K) required to observe QAHE[61-64,67] remains a major barrier for applicability of this exotic effect. Yet, it is believed that achieving a higher Curie temperature, along with stronger long-range ferromagnetic coupling, could result in the observation of QAHE at much higher temperatures. To this end, MBE-grown TI films has fundamental advantages over bulk crystals: for example, the Curie temperature ($T_c$) was increased from 20 K for bulk crystals to 177 K for Cr-doped $Sb_2Te_3$ thin films, and from 24 K for bulk crystals to 190K for V-doped $Sb_2Te_3$ thin films[214,215]. The higher Curie temperatures of the MBE-grown thin films may result from increased magnetic dopant solubility at the lower temperatures used for MBE growth than the bulk crystals. However, while increased magnetic doping may raise $T_c$ and widen the magnetic exchange gap, doing so degrades the sample's crystallinity and eventually destroys TSS, such that either the bulk becomes topologically trivial or the impurity concentration exceeds the solid solubility of the host material. Table II summarizes many examples of magnetic doping in TI thin films or single crystals.

**Table II: Key properties of magnetically-doped TI thin films or single crystals**

| Material | Magnetic dopant | Form | Ref. | Comment |
|---|---|---|---|---|
| $Sb_2Te_3$ | V | Bulk | [214] | $T_c \sim$ 24 K for $Sb_{1.97}V_{0.03}Te_3$ |
| $Sb_2Te_3$ | Cr | Bulk | [215] | $T_c \sim$ 20 K for $Sb_{1.905}Cr_{0.095}Te_3$ |
| $Sb_2Te_3$ | Mn | Bulk | [216] | $T_c \sim$ 17 K for $Sb_{1.985}Mn_{0.015}Te_3$ (1.5% substitutional doping) |
| $Sb_2Te_3$ | Cr & V | Bulk | [217] | $T_c$ of $Sb_{1.98-x}V_{0.02}Cr_x Te_3$ samples comparable to that of $Sb_{1.98}V_{0.02}Te_3$; yet, significant enhancement in the remanent magnetization when Cr is co-doped |
| $Sb_2Te_3$ | Mn & V | Bulk | [217] | Adding Mn to $Sb_{1.984}V_{0.016}Te_3$ decreased $T_c$ and at higher concentration suppressed ferromagnetism likely due to antiferromagnetically coupled Mn-ion pairs |
| $Sb_2Te_3$ | V | Thin film | [218] | $T_c \sim$ 177 K for $Sb_{1.65}V_{0.35}Te_3$ |
| $Sb_2Te_3$ | Cr | Thin film on $Al_2O_3$(0001) | [219] | $T_c \sim$ 190 K for $Sb_{1.41}Cr_{0.59}Te_3$ |
| $Sb_2Te_3$ | Cr | Thin film on $SrTiO_3$(111) | [220] | Field effect modulation of AH loop in $Sb_{2-x}Cr_xTe_3$; $T_C \sim$ 55 K for 10 QL $Sb_{1.7}Cr_{0.3}Te_3$ films |
| $Sb_2Te_3$ | Cr | Thin film on $Al_2O_3$(0001) | [221] | Highly crystalline up to x = 0.42 (in $Cr_xSb_{2-x}Te_3$); $T_c \sim$ 125 K for a 60 QL-thick $Cr_{0.42}Sb_{1.58}Te_3$ with $n_{sheet}$ = $8.6 \times 10^{13}$ cm$^{-2}$ at 1.5 K |
| $Bi_2Te_3$ | Fe | Bulk | [222] | $T_c \sim$ 12 K for $Bi_{1.92}Fe_{0.08}Te_3$ |
| $Bi_2Te_3$ | Mn | Bulk | [216] | $T_c \sim$ 10 K for $Bi_{1.98}Mn_{0.02}Te_3$ |
| $Bi_2Te_3$ | Mn | Thin film on $SrTiO_3$(111) | [223] | Skyrmion-induced topological Hall effect in $(Bi_{0.9}Mn_{0.1})_2Te_3$ films by varying the film thickness below $T_c \sim$ 18 K |
| $Bi_2Te_3$ | Mn | Thin film on InP(111)A | [224] | AHE observed in n-type Mn-doped $Bi_2Te_3$ films below $T_c \sim$ 17 K |
| $Bi_2Se_3$ | Fe | Bulk | [222] | Paramagnetic |
| $Bi_2Se_3$ | Mn | Bulk | [225] | Spin glass behavior with blocking $T_c \sim$ 32 K for $Bi_{1.97}Mn_{0.03}Se_3$ |

| Bi$_2$Se$_3$ | Cr | Thin film on Si (111) | 226 | $T_c \sim$ 20 K for 5.2% of Cr; for higher Cr content, $T_c$ drops with deteriorating crystallinity |
|---|---|---|---|---|
| Bi$_2$Se$_3$ | Cr | Thin film on Si (111) | 227 | $T_c \sim$ 30 K for Bi$_{1.94}$Cr$_{0.06}$Se$_3$, no hysteresis loop |
| Bi$_2$Se$_3$ | V | Thin film on SrTiO$_3$(111) | 195 | $T_c \sim$ 16 K for 7 QL-thick Bi$_{1.88}$V$_{0.12}$Te$_3$ |
| Bi$_x$Sb$_{2-x}$Te$_3$ | Cr | Thin film on SrTiO$_3$(111) | 61 | $T_c \sim$ 16 K for 5 QL-thick Cr$_{0.15}$(Bi$_{0.1}$Sb$_{0.9}$)$_{1.85}$Te$_3$; QAH observed. |
| Bi$_x$Sb$_{2-x}$Te$_3$ | V | Thin film on SrTiO$_3$(111) | 62 | $T_c \sim$ 35 K for 4 QL-thick (Bi$_{0.29}$Sb$_{0.71}$)$_{1.89}$V$_{0.11}$Te$_3$; QAH observed. |
| Bi$_x$Sb$_{2-x}$Te$_3$ | Cr & V | Thin film on SrTiO$_3$(111) | 211 | $T_c \sim$ 25 K for 5 QL-thick (Cr$_{0.16}$V$_{0.84}$)$_{019}$(Bi$_x$Sb$_{1-x}$)$_{1.81}$Te$_3$; enhanced QAH as a result of co-doping |

As an alternative to magnetic doping, bismuth telluride combined with manganese telluride layers was recently reported to be the first intrinsic antiferromagnetic topological insulator (AFMTI)[228,229]. The stoichiometric compound, MnBi$_2$Te$_4$, grows in septuple layers (SL) in the sequence, Te-Bi-Te-Mn-Te-Bi-Te (in MBE, the SL structure can be achieved either thermodynamically or kinetically by alternate growth of a Bi$_2$Te$_3$ quintuple layer and a MnTe bilayer[228,230,231]). Thin AFMTI films are predicted to be axion insulators for even SL films (so that the top- and bottom-most Mn planes have antiparallel magnetization) and QAH insulators for odd SL films (so that the top- and bottom-most Mn planes have parallel magnetization). This prediction is already approximately (under finite magnetic field) confirmed in thin flakes[75,232]. On the other hand, AFMTI thin films (grown by MBE) have not yet reached the quantized regime, but with further materials development, MnBi$_2$Te$_4$ thin films could be better than the Cr-(Bi$_x$Sb$_{1-x}$)$_2$Te$_3$ system because MnBi$_2$Te$_4$ can, in principle, be fully ordered, whereas CrBiSbTe is intrinsically disordered due to the random doping process.

Another way to avoid magnetic doping is to introduce magnetism into TI by proximity effect. In this scheme, a nonmagnetic TI layer is grown on top of an insulating magnetic layer (or vice versa). Inducing magnetism by proximity allows independent optimization of the magnetic and electronic properties of the system. In particular, a high T$_c$ magnetic insulator may be selected without concern of dopant solubility or dopant-induced disorder and scattering in the TI layer. The ideal magnet should be an insulator, so that conduction through the magnet

does not eclipse conduction in the TI, and should have perpendicular magnetic anisotropy, so that a Dirac mass is induced in the TI. Rather counter-intuitively, it may not be crucial for the magnetic insulator to be a ferromagnet: since a TI on a magnetic insulator is primarily coupled to the top layer of the magnet, an antiferromagnet (AFM) could induce ferromagnetism in the TI interfacial layer, without producing stray fields or affecting the properties of the neighboring TI layer due to AFM's nearly zero net magnetization[233].

Much work has sought a high $T_c$, out-of-plane magnet capable of proximitizing an exchange gap in a TI layer (although, in a special case, an in-plane magnet could, theoretically, also induce a QAHE[234]). Candidate magnetic layers include the ferromagnetic insulator EuS[99-101,235,236], ferrimagnet insulator $Y_3Fe_5O_{12}$(YIG)[102-105,237] with in-plane magnetization and $Tm_3Fe_5O_{12}$ (TIG, $T_c = 560$ K) with out-of-plane magnetization[106], antiferromagnetic conductor CrSb[107,238] with Néel temperature ($T_N$) ~700 K, ferromagnetic insulator $BaFe_{12}O_{19}$ with $T_c$ ~723K[108], and ferromagnetic insulator $Cr_2Ge_2Te_6$ (CGT)[109,239]. Although a QAHE has not been observed yet, an anomalous Hall effect was observed up to 400 K in a bilayer film of $Bi_xSb_{2-x}Te_3$ and the ferrimagnetic insulator TIG, grown on a (111)-oriented substituted gadolinium gallium garnet (SGGG) substrate[106].

In another study, Cr dopants in a magnetic topological insulator (MTI) experience interfacial exchange coupling, once put in contact with an almost lattice-matched AFM transition-metal pnictide CrSb. This is evidenced by enhancement in magnetization loop where coercive field increases by 67 mT for MTI/AFM bilayer and by 90 mT for trilayer AFM/MTI/AFM. Interfacial exchange coupling in the heterostructure tailors the spin texture in both the AFM and MTI layers, introducing an effective long-range exchange coupling between the MTI layers mediated by the AFM layers. However, unlike TIG, the hysteresis loop persisted only up to 80 K and CrSb is not insulating.

When coupled with $Bi_2Te_3$, $T_c$ of the out-of-plane magnet $Cr_2Ge_2Te_6$ (CGT) is enhanced from 61 K to 108 K, accompanied by an anomalous Hall effect[109]. Even more dramatically, $Bi_2Se_3$/EuS bilayers have exhibited interfacial magnetism even at room temperature, as confirmed by polarized neutron reflectometry (PNR), despite bulk EuS having $T_c = 17$ K[240]. This implies that electronic interaction between the Eu atoms and the TI surface states enhances magnetic order in both materials, causing high temperature magnetization in the 2 QL-thick TI layer. Furthermore, proximity to the TI layer changes the magnetic anisotropy of EuS. While EuS films favor in-plane magnetization, when coupled to a thin TI layer, SOC leads to an out-of-plane magnetic moment in the TI surface.

Despite these interesting progresses in TI/magnet heterostructures, the main problem in this approach is that the exchange coupling is too weak to realize QAHE. Considering that exchange coupling in insulating layers is limited to sub-nanometer scale, the interface quality should be even more critical for achieving QAHE via proximity effect than for the non-magnetic TI thin films as reviewed in the previous sections. So far the highest anomalous Hall signal relying purely on proximity effect is achieved in BST films grown on chemically and structurally matched CGT substrate[239]. Although BST thin films sandwiched by $Zn_{1-x}Cr_xTe$ layers claimed observation of QAHE by proximity effect[241], considering that the observed QAHE is very similar to that of Cr-doped BST system, one cannot rule out the possibility of Cr diffusion into the BST film during the film growth. Anyhow, the very question of whether we can enhance the temperature of QAHE by proximity effect still remains elusive, but if that is to be ever possible, exquisite interface engineering should be a must.

IV.c    Solution to counter-doping problems of $Bi_2Se_3$ and $Sb_2Te_3$ thin films via interface engineering

As mentioned before, although p-type bulk $Bi_2Se_3$ crystals were achieved quite early through Ca counter-doping, making p-type $Bi_2Se_3$ thin films remained challenging. One possible conjecture is that this discrepancy could originate from the low temperature growth of thin films where the dopants may not become activated (in fact, annealing helped activate p-type dopants in the case of GaN-based blue LEDs)[242]. However, even p-type $Bi_2Se_3$ bulk crystals convert to n-type once cleaved into thin flakes [243]. This shows that the inability to p-dope $Bi_2Se_3$ thin films is more due to the sample's thinness rather than the growth condition. In fact, other than a simple graph in ref. [180] without any other transport data, there was not any report of p-type $Bi_2Se_3$ thin ($< 100$ nm) films until 2018 and reduced carrier density was the only success[19, 63]. The only p-type $Bi_2Se_3$ film with clear Hall effect data before 2018 was by Sharma *et. al*[244] on a thick (256 nm) film, which is ion-implanted by Ca, followed by annealing.

As it turns out, counter-doping of a TI system is much more challenging than that of a regular semiconductor because of the surface states: counter-doping of bulk and surface defects must be considered separately. In general, because of lower coordination numbers, the surface tends to have more defects than the bulk. Accordingly, due to the large surface to bulk ratio, the defect density in TI thin films can easily be dominated by the surface defects: unlike in semiconductors, most of these defects contribute mobile carriers due to the presence of topological surface states. The problem is that this surface defect density can be easily much higher than the solubility limit of compensation dopants. For example, the interfacial defect density of $Bi_2Se_3$ thin films grown on commercial substrates is so high that it goes beyond the solubility limit of Ca doping, failing to convert n-type conductance into p-type up to the maximum counter-doping. Only after the interfacial defect density was substantially reduced with the use of the BIS (($Bi$,$In$)$_2Se_3$) buffer and proper capping layers, it was possible to reach the p-regime of $Bi_2Se_3$ thin films with Ca counter-doping[59]. It turns out that previous failures of counter-doping for $Bi_2Se_3$ thin films were because of: first, neutralization/oxidation of the

counter-dopants due to air exposure and second, high density of interfacial defects due to chemical and structural mismatch with the substrate.

Moon *et al.*[59] showed that, as the Ca doping level in buffer-layer-based $Bi_2Se_3$ films capped with $MoO_3$/Se increases, it compensates for the intrinsic n-type carriers and the (n-type) sheet carrier density gradually decreases (Fig. 7). Upon adding more Ca, the slope of $R_{xy}(H)$ curve changes from negative (n-type) to positive (p-type), passing through a non-linear n-p mixed regime (Fig.7a). As the films get thinner, higher level of Ca doping is required to reach the n-p mixed or p regimes, suggesting that the relative density of interfacial defects with respect to bulk defects grows as the film gets thinner, even in these interface-engineered films. Nonetheless, p-type $Bi_2Se_3$ films are achieved for all thickness range from 50 QL down to 6 QL. For thick films, as long as they are capped, p-type samples can be achieved for both with and without the buffer-layer. The films become n-type again with further increase in Ca doping, suggesting  that, beyond a certain concentration, the compensation dopants start to act as n instead of p-type dopants, except for 6 QL-thick sample, which degrades and becomes insulating probably due to a disorder-driven topological phase transition[245].

Furthermore, pure p-regime shows higher carrier density than the pure n-regime (lowest p-type sheet carrier density of $\sim 1.5 \times 10^{12}$ $cm^{-2}$ vs. lowest n-type carrier density of $\sim 6.0 \times 10^{11}$ $cm^{-2}$). Also, the mobility sharply decreases as soon as the majority carrier type changes from n to p-type, which is very likely due to the nature of $Bi_2Se_3$ band structure, where the Dirac point is very close to the bulk valence band and the surface band broadens noticeably on the p-side.

Moreover, as shown in Fig. 7, the Hall resistance at high magnetic fields is perfectly quantized at $R_{xy} = \frac{h}{e^2}$ with a vanishing sheet resistance on the n-side up to 0.08% of Ca, indicating the emergence of a well-defined, chiral edge channel at high magnetic fields. With small increase in Ca doping (0.09%), the sheet magnetoresistance soars from close-to-zero to

a large insulating value at high magnetic fields, which indicates formation of a gap at the zeroth LL at high magnetic fields, thereby leading to vanishing edge channel and insulating sheet resistance: accessing the details of this zeroth LL was eventually made possible by the interface-engineered ultralow carrier density $Sb_2Te_3$ films[60] as discussed in the previous section IV.a. With further increase of Ca doping, as the majority carrier type changes to p-type, the signature of QHE significantly degrades, which is also consistent with the absence of Landau levels on the p-side of $Bi_2Se_3$ as measured by previous scanning tunneling spectroscopy (STS) studies[20,23].

In a follow-up study, Moon *et al* were able to achieve p-type $Bi_2Se_3$ by doping interface-engineered $Bi_2Se_3$ films with Pb[246]. Prior to this work, obtaining p-type with Pb doping was not possible in conventional thin films or even bulk crystals. Moreover, unlike doping with lighter elements such as Ca, doping with heavy element like Pb does not weaken the SOC strength.

Similarly, Ti-doping was never successful in converting p-type $Sb_2Te_3$ to n-type and at best it resulted in reduced carrier density[247-249]. It was found that p- to n-type conversion becomes possible only in interface-engineered films with proper buffer and capping layers, which once again shows the importance of suppressing both surface and bulk defects in TI thin films[60,250].

IV.d     Quantized Faraday and Kerr rotation and the evidence for axion electrodynamics

Time- domain THz spectroscopy (TDTS) is a non-destructive technique which provides useful information about the electrodynamics of TIs[33,34,76,77,152,163,251-255]. As a complement to DC transport measurements, data such as carrier density and mobility of a TI system can be extracted from the measured complex transmission/conductivity. Furthermore, time-domain magneto-terahertz spectroscopy can be used to distinguish the bulk/2DEG and topological surface state contributions and to extract cyclotron resonances from TSS in bulk-insulating TIs. It can also be exploited to study quantum phase transition between a topological insulator and

a normal insulator. More importantly, with the aid of high-precision time-domain terahertz polarimetry technique on low-Fermi-level bulk insulating TIs, the topological magneto-electric effect can be probed[76,77,256,257].

In principle, true TI is considered as a bulk magnetoelectric material whose magnetoelectric response is a quantized coefficient and its size is set by the fine-structure constant $\alpha = \frac{e^2}{2\varepsilon_0 hc}$. Nonetheless, just as any quantized phenomena, in order to observe the quantized effect, the relevant quantum number should be low enough: this requires not only insulating bulk but also the Fermi level being close to the Dirac point. This milestone was finally achieved with the interface-engineered bulk-insulating $Bi_2Se_3$ films capped by $MoO_3$/Se (capping layers are transparent in THz regime) whose surface $E_F$ is only about 30 ~ 60 meV from the Dirac point. On these samples, by utilizing time-domain terahertz polarimetry, Wu *et al* have finally observed the first signature of axion electrodynamics in terms of quantized Faraday and Kerr rotation[76]: Fig. 8. The consequences of axion electrodynamics are the additional source and current terms in the modified Gauss's and Ampère's laws, which are responsible for a half-integer QHE on the TI surface. The Faraday and Kerr rotations in quantum regime are formulated as:

$$\text{Faraday rotation: } \tan(\phi_F) = \frac{2\alpha}{1+n}\left(N_t + \frac{1}{2} + N_b + \frac{1}{2}\right)$$

$$\text{Kerr rotation: } \tan(\phi_K) = \frac{4n\alpha}{n^2-1}\left(N_t + \frac{1}{2} + N_b + \frac{1}{2}\right),$$

where $\alpha$ is the fine structure constant, $n \approx 3.1$ is the index of refraction for the sapphire substrate, and $N_b$ and $N_t$ are the highest filled LL of the top and bottom surfaces of the film, which depend on the Fermi level and magnetic field. Figure 8 shows the quantized Faraday and Kerr rotations for different sample thicknesses: thicker sample has slightly higher carrier density and thus higher filling factor. Notably, magnetic field of 7 T is large enough to induce the quantization,

which is much lower than the magnetic field (~25 T) required for QHE in $Bi_2Se_3$ samples with DC transport. This is most likely because THz measurement only detects the surface and the bulk and does not probe the edges, which, in the case of TIs, harbors non-chiral edge states except for the zeroth Landau level[60]. The required field for quantized Faraday/Kerr rotation is even smaller for the ultra-low-carrier density $Sb_2Te_3$ films, which also exhibit quantized Faraday/Kerr rotations for both n- and p-regimes[258].

IV.e Finite-size effect in topological phase transitions of interface-engineered $(Bi_{1-x}In_x)_2Se_3$ films

In an infinite-size TI, if its SOC strength is gradually reduced, the TI eventually transforms into a trivial insulator beyond a critical point of SOC, at which point the bulk gap closes[78,162-165,259,260]. However, Salehi *et al* have shown that, by utilizing topologically-tunable, interface-engineered $(Bi_{1-x}In_x)_2Se_3$ thin films augmented by theoretical simulations [78], this conventional picture of topological phase transition (TPT), envisioned from infinite-size samples, has to be substantially modified for finite-thickness samples due to quantum confinement and surface hybridization: Fig. 9.

When an infinite-size system undergoes TPT, at a critical strength of SOC the bulk gap closes and reopens, and the topological surface states disappear (by merging into the bulk states): SOC is controlled by In concentration in $(Bi_{1-x}In_x)_2Se_3$[162,164,165]. In contrast, for finite-thickness system, the bulk band gap never closes completely due to diverging hybridization effect of TSS at the critical point[78]: Fig. 9i and k. Actually, the surface hybridization effect of TSS was first observed (with ARPES studies) on regular $Bi_2Se_3$ films, which revealed that the top and bottom surface states hybridize and form a gap at the Dirac point below 6 QL of thickness[95,146]. Similar studies are also done for $Bi_2Te_3$ and $Sb_2Te_3$ (using scanning tunneling spectroscopy) thin films, revealing that the hybridization occurs at 2 QL[147] and 3 QL[145], respectively. In $(Bi_{1-x}In_x)_2Se_3$ thin films, however, the effective thickness of TSS diverges at

the critical concentration of In, so the hybridization effect should open a gap at the Dirac point for any finite-thickness films at the critical point.

In an ideal TI sample with its Fermi level at the Dirac point, as soon as the surface gap opens at the Dirac point, the material should undergo metal-to-insulator transition. So, in principle, this TPT and the related finite-size effect can be detected even with transport measurements. However, high Fermi levels of early generation TI films made this task practically impossible, because these samples remained metallic throughout the transition[162]. It was only after the development of the interface-engineered $(Bi_{1-x}In_x)_2Se_3$ films with low Fermi levels (few tens of meV from the Dirac point) that it was made possible to detect TPT-driven MIT and its finite-size effect using transport measurements[78] (Figs.9a to h). This study shows that topological phase transition in finite-thickness TI films goes through two separate quantum phase transitions: first, topological-metal to normal-metal and then, normal-metal to normal-insulator, generating a well-defined thickness-dependent phase diagram as shown in Fig. 9j and k.

# V.    Conclusion and outlook

-All aforementioned experiments corroborate the important role of (interfacial) defect suppression, through various growth methodologies, in solving the bulk conduction problem of thin film TIs and eventually reaching quantum regime of TSS. Although TI thin films can grow on almost any substrates regardless of lattice matching due to the van der Waals bonding nature between layers, the very presence of TSS allows charge defects to easily donate mobile carriers without trapping. This is rather the opposite to the situation of conventional semiconductors, where charge defects tend to trap, rather than mobilize, carriers, particularly at the interfaces. Accordingly, in order to reach the quantum regime of TSS near the Dirac point, it is critical to suppress charge defects. The far superior qualities of $Bi_2Se_3$ films grown on $(Bi_{1-x}In_x)_2Se_3$ buffer layers and bulk crystals[261] compared to those on lattice-constant-

matched, yet chemically and structurally ill-matched, InP substrate, clearly showcases the importance of chemical and structural matching for reaching the quantum regime of TSS in TI films.

- It is notable that QHE requires even lower Fermi level than QAHE in TI materials. As a matter of fact, QAHE, which had never been observed in any other systems before, was observed ahead of QHE in TI systems, despite QHE having been observed in various 2D systems for many decades before[262]. Specifically, QAHE was observed in magnetic TI thin films in 2013[61], but the first QHE in a TI system was observed only in 2014 in bulk flakes[68] and in 2015 in thin film TIs[57,69]. The exact reason why QAH samples can tolerate higher residual carrier densities than do QH TI samples is unknown yet, but it may be because the internal magnetic strength provided by exchange coupling in a magnetic TI is much stronger than can be provided by any external magnetic field.

-Although QHE are now observed in multiple TI platforms including, gated BSTS single crystals[68,70], gated MBE-grown BST films[69], and interface-engineered MBE-grown pure binary $Bi_2Se_3$ films both with and without gating[57,263], it was only ultra-low carrier density interface-engineered Ti-doped $Sb_2Te_3$ films which allowed access to the details of the zeroth LL of TSS. These extreme quantum-regime films revealed QIT (quantum Hall to insulator transition) at high magnetic fields[60]. Although QIT implies that the topological (Chern) number changes from 1 to 0, this does not *ipso facto* imply that every QIT should belong to the same universality class. In fact, the extracted dynamical critical exponent $z \approx 2$ in TI $Sb_2Te_3$ films is clearly different from $z \approx 1$[264,265] of 2DEGs, suggesting that QITs in these two systems should belong to different universality classes. Whether this is due to the different band structures in TIs vs. 2DEGs, or due to some other higher order effects is an open question. Similar studies with other Dirac systems, such as graphene could further shed light on this question.

-While interface engineering schemes have been critical to reaching the quantum regime of topological surface states in non-magnetic TIs by suppressing the residual sheet carrier densities from $\sim 3 \times 10^{13}$ cm$^{-2}$ down to $\sim 1 \times 10^{11}$ cm$^{-2}$, similar level of interface-engineering schemes have not been fully exploited yet for QAHE. Nonetheless, the enhancement of QAH signatures in magnetic modulation-doped and co-doped MTI films does show that defect control is also important for QAHE. We have yet to see if more exquisite interface engineering schemes can further boost the temperature required for QAHE.

In order to boost the temperature for QAHE, magnetic order in TIs must be better understood. In conventional diluted magnetic semiconductors, long-range ferromagnetic order is believed to arise through coupling between distant magnetic impurities mediated by bulk itinerant charge carriers. However, QAHE exists in an insulating regime ($E_F$ sits in not only in the bulk bandgap, but also in the magnetic exchange-induced TSS gap). The magnetism of QAH insulators, therefore, cannot be attributed to the Ruderman-Kittel-Kasuya-Yosida (RKKY) mechanism, which requires bulk itinerant carriers. Experimental evidence supports this claim. For example, Chang $et\ al.$ [266] show that the Curie temperature is almost independent of the carrier density, and they argue that the sheet carrier density of $4 \times 10^{12}$ cm$^{-2}$ (extracted from the Hall slope) is too small to support a RKKY mechanism. It is therefore believed that ferromagnetism for the QAH insulators emerges from the van Vleck mechanism. In any case, further investigation is required to pinpoint the exact mechanism of ferromagnetism and the role of surface and bulk carriers.

-Another notable feature of AHE/QAHE in MTI is that its sign is almost exclusively determined by the magnetic ion but not by the carrier type. (Sb,Bi)$_2$Te$_3$ thin films doped with Cr[266] (Fig. 6b and c) or V[62] feature positive AH loops regardless of whether the majority carrier type is n- or p-type. Also, the observed QAH loop for Cr-[61] and V-doped (Sb,Bi)$_2$Te$_3$[62] films

has always a positive sign and even at elevated temperatures, when the system is far from the QAHE regime, the sign of the loop remains unchanged (Fig. 6d). Similarly, only positive AHE hysteresis loops have been observed also in V-doped $Bi_2Se_3$ films [195] as well Cr modulation-doped $Bi_2Se_3$ films[196], even if Cr-doped $Bi_2Se_3$ films exhibit only negative-slope paramagnetic AHE (without any hysteresis loop)[197]. On the other hand, Mn-based MTIs[267] always exhibit negative AHE loops (Fig. 6a). Checkelskey *et al.* also showed that the AH loop for these samples disappears when $E_F$ locates deep in the bulk conduction band ($n_{2D} \approx 3 \times 10^{13}$ cm$^{-2}$), indicating that the ferromagnetism is not mediated by bulk carriers. Quantized AHE recently reported in flakes of $MnBi_2Te_4$ crystals also exhibit only negative AHE loops[75,232]. There are also other MTI system such as $(Sb,Bi)_2Te_3$/TIG proximity structure[106], that exhibit only negative AH loops.

Magnetism, either via doping or proximity, induces an exchange gap as well as a finite Berry curvature in an MTI. All the above examples show that the AHE/QAHE response depends not on the carrier type of the MTI system but on the Berry curvature of energy bands (Berry curvature can be thought of as a magnetic field in k-space) and on microscopic origin of the magnetism-induced exchange gap. This is because the Hall conductance can be written as $\sigma_{xy} = \frac{e^2}{2\pi h} \int_{occ} \mathcal{B} d^2 k = \frac{e^2}{h} C$ (i.e. the integral of Berry curvature $\mathcal{B}$ over the entire occupied states; $C$ is the Chern number, which is $\pm 1$ in TIs). Moreover, if the integration covers the entire Brillouin-zone, which is the case for QAHE where the Fermi level is located in the exchange gap of an MTI, the Hall conductance becomes an integer number: see Fig. 6e and f. However, in the case of AHE in an MTI or in a normal FM metal[268], where the band is partially filled, the integration is not over the entire Brillouin zone, and thus, the Hall resistance/conductance is not quantized. Hence, how magnetism couples with an MTI and how it induces an exchange gap is what could determine the sign of the loop.

- Combination of superconductors and TIs[269-271] can lead to another set of interesting physics. For instance, it is believed that proximity effect between a TI and an s-wave superconductor could generate Majorana fermions at vortices with the interface hosting spinless p-wave superconductivity. Additionally, in principle, a superconductor in good electrical contact with a QAH insulator should proximitize a superconducting gap in the QAH insulator, forming a topological superconductor (TSC) with chiral Majorana edge modes (CMEMs) propagating along the boundary of the TSC. Under an appropriate applied field, it is predicted that only a single CMEM propagates, which leads to a half-quantized two-terminal conductance ($\sigma_{12} = \frac{e^2}{2h}$)[272]. Such a half-quantized conductance plateau was reported a few years ago[79] but recently refuted by other group[80]. In fact, a half-quantized conductance plateau could instead result from dissipative effects in the QAH insulator, rather than the presence of CMEMs[273,274]. Considering that interface is where all the interesting physics occurs in these topological materials, finding proper interface engineering schemes must be essential in order to reveal all these intriguing topological features.

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

**Figure 1: Interface-engineered Bi₂Se₃ and Sb₂Te₃ films with ultralow Fermi levels**

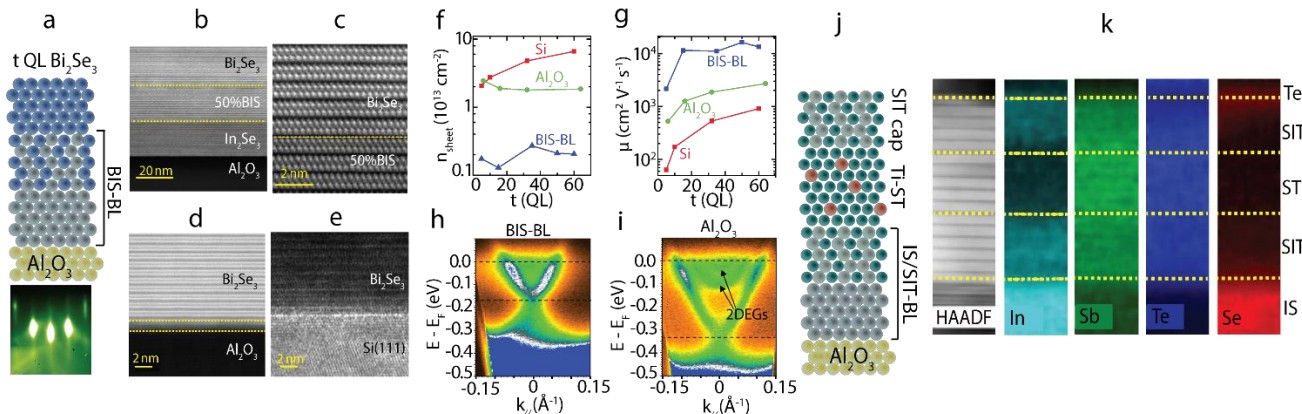

**a,** Cartoon of a buffer-layer-based Bi₂Se₃ film where the buffer-layer (BIS-BL) is composed of a 20 QL-thick In₂Se₃ and a 20 QL-thick (Bi₀.₅In₀.₅)₂Se₃ (for growth details see ref. [57]). The RHEED image of Bi₂Se₃ film (bottom panel) shows high quality 2D growth. **b,** and **c,** High angle annular dark-field scanning tunneling electron microscopy (HAADF-STEM) image of Bi₂Se₃ grown on BIS-BL shows an atomically-sharp interface between Bi₂Se₃ and BIS-BL (50%BIS is a short form of (Bi₀.₅In₀.₅)₂Se₃), while **d,** Bi₂Se₃ grown directly on Al₂O₃ has clearly disordered interface. **e,** TEM image of Bi₂Se₃ grown on Si(111) (from ref. [40]) also shows a hazy interface. Comparison of **f,** sheet carrier densities ($n_{sheet}$) and **g,** Hall mobilities ($\mu$) of Bi₂Se₃ films grown on BIS-BL, Al₂O₃ (0001) and Si (111) for different film thicknesses. ARPES of Bi₂Se₃ grown on **h,** BIS-BL shows the bulk-insulating nature of the film with the surface Fermi level in the bulk gap and **i,** Al₂O₃(0001), where the surface Fermi level is located above the bottom of the conduction band and the 2DEG originating from downward band bending is marked. (a), (b), (c), (d), (f), (g), (h), and (i) are adapted from ref. [57]. **j,** Schematic of the buffer-layer-based Sb₂Te₃ film structure. The buffer-layer (IS/SIT-BL) consists of 20 QL In₂Se₃ and 15 QL-thick (Sb₀.₆₅In₀.₃₅)₂Te₃ BL as a template for the successive layer of ultra-low carrier density Ti-doped Sb₂Te₃. The film is capped by 15 QL (Sb₀.₆₅In₀.₃₅)₂Te₃ providing a symmetric condition for top and bottom surfaces of the Sb₂Te₃ film. **k,** The first panel shows HAADF-STEM for In₂Se₃ /5 QL (Sb₀.₆₅In₀.₃₅)₂Te₃/5 QL Sb₂Te₃/5 QL (Sb₀.₆₅In₀.₃₅)₂Te₃ with an additional 10 nm Te capping for further protection against STEM sample preparation processes. Yellow dashed lines mark each interface. The rightward four panels show the elemental mapping electron dispersive X-ray spectroscopy (EDS) images for In (light blue), Sb (green), Te (dark blue), and Se (red). If a specific element is present in a layer, then the corresponding color appears boldly in that section; in its absence, the layer appears dark. Figures (j) and (k) are adapted from ref. [60]

**Figure 2: QHE in flakes of TI single crystals**

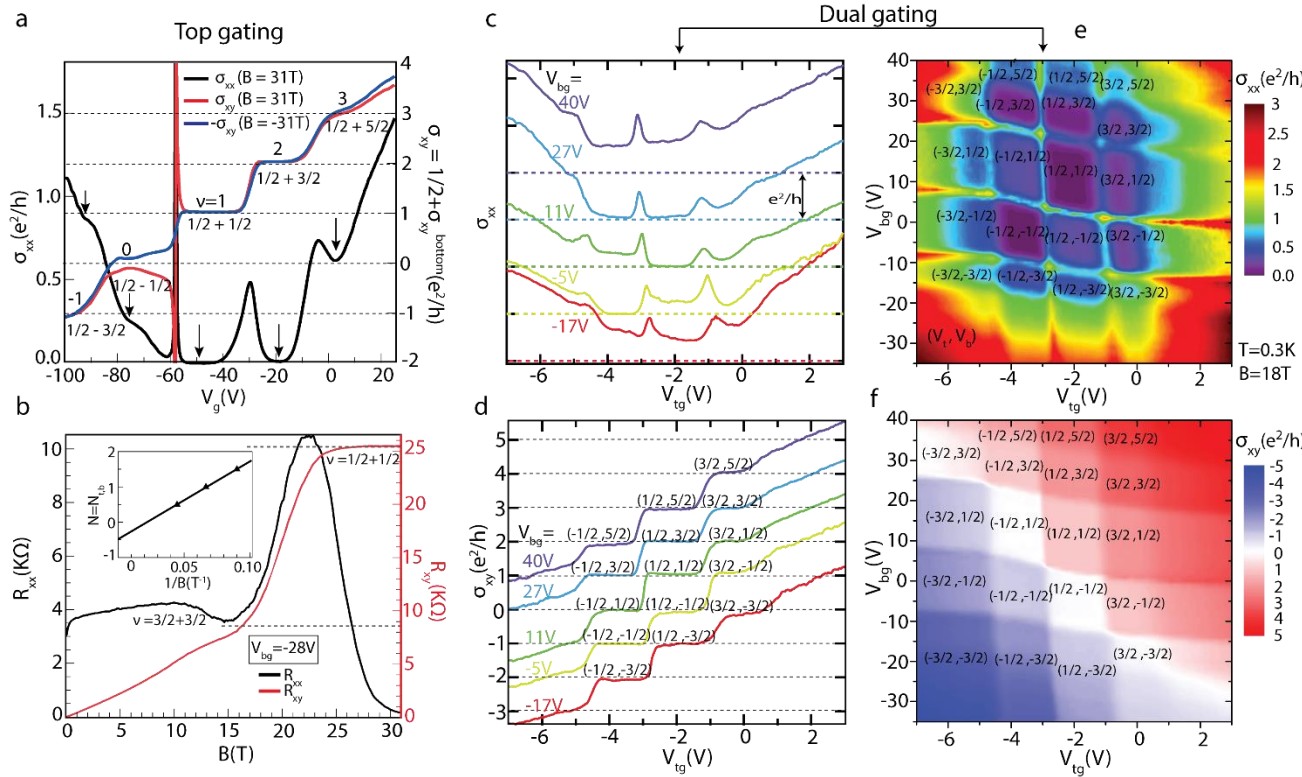

**a,** Gate voltage ($V_g$) dependence of the 2D longitudinal conductivity ($\sigma_{xx}$) and Hall conductivity ($\sigma_{xy}$) for a back-gated 160-nm-thick exfoliated BSTS flake on SiO₂/Si substrate and at magnetic field 31T. This experiment is the first-time observation of QHE in a TI system. $\nu e^2/h$ plateaux observed in $\sigma_{xy}$ are marked with the corresponding top surface ($\nu_{top}$) + bottom surface ($\nu_{bot}$) Landau filling factors of the quantum Hall states. Arrows show the corresponding $\sigma_{xx}$ minima. **b,** Magnetic field dependence of $R_{xx}$ and $R_{xy}$ measured at $V_g = -28$ V and at 350 mK on a different sample with the same thickness. $R_{xy}$ shows clear $\nu = 1$ plateau (accompanied by an almost vanishing $R_{xx}$) and an almost $\nu = 3$ plateau (accompanied by an $R_{xx}$ minimum). The inset shows the LL fan diagram with 1/2-intercept in LL index, which is a hallmark of Dirac fermions and underlies the half-integer QHE. Figures (a) and (b) are adapted from ref. [57] **c,** $\sigma_{xx}$ and **d,** $\sigma_{xy}$ (shifted vertically by $\frac{e^2}{h}$ steps for clarity) as functions of $V_{tg}$ and for 5 different values of $V_{bg}$ at magnetic field 18 T and at 300 mK in a ~100 nm-thick exfoliated BSTS flake on an SiO₂/Si substrate for bottom-gate and with 40 nm-thick h-BN as a top-gate dielectric, which features a series of ambipolar two-component half-integer Dirac quantum Hall states. Quantum Hall states are labelled by the corresponding (top, bottom) surface filling factors. **e,** 2D color maps of $\sigma_{xx}$ (top panel) and (bottom panel) as functions of $V_{tg}$ and $V_{bg}$ at 18 T and 300 mK. Figures (c), (d), and (e) are adapted from ref. [70]

## Figure 3: QHE in MBE-grown TI films

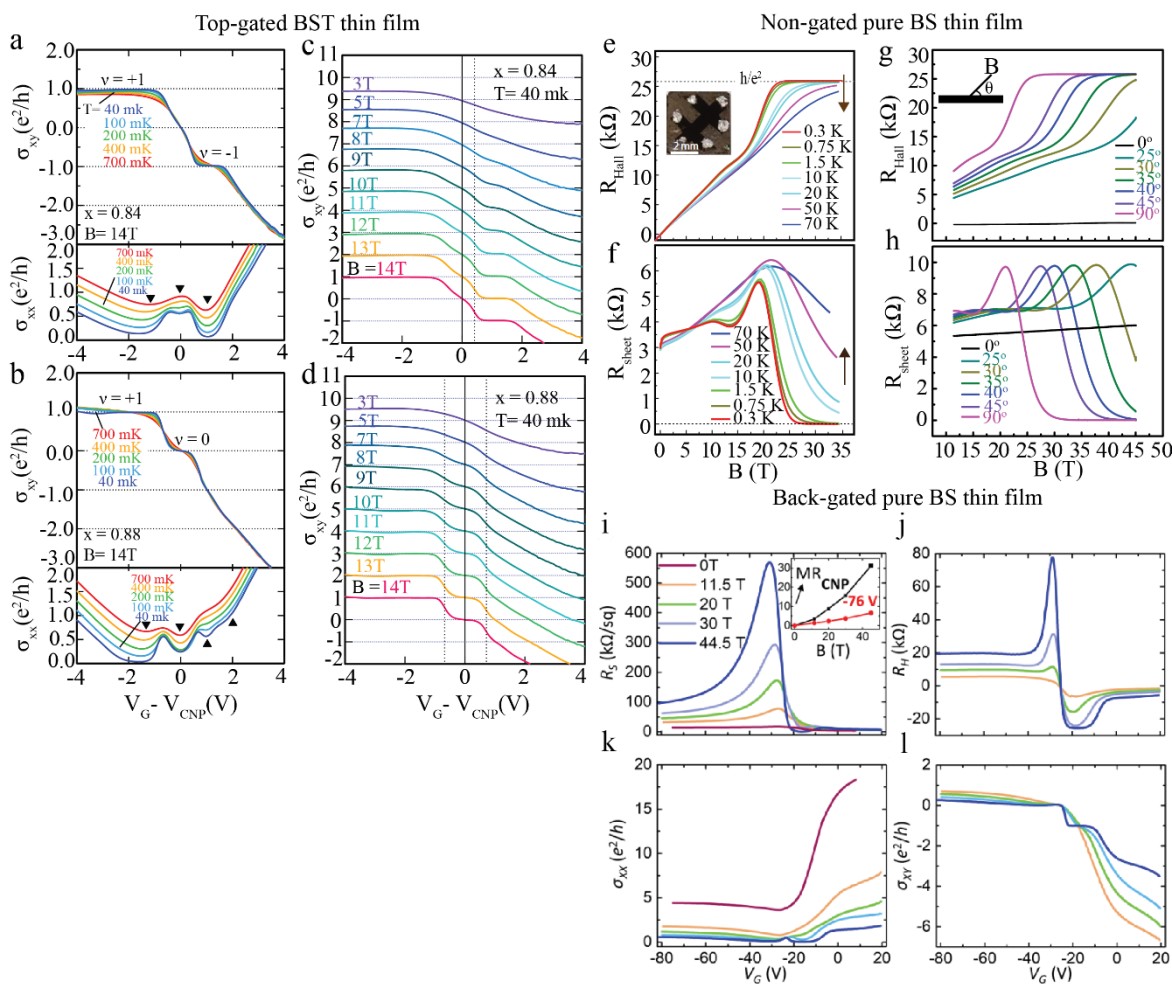

Gate voltage ($V_G$-$V_{CNP}$) dependence of extracted 2D $\sigma_{xx}$ and $\sigma_{xy}$ at various temperatures from 40 to 700 mK in magnetic field of 14 T for a top-gated MBE-grown 8 QL-thick film of **a,** $(Bi_{0.16}Sb_{0.84})_2Te_3$ where LLs of top and bottom surface are degenerate and $v = \pm 1$ plateaux are measured and **b,** $(Bi_{0.16}Sb_{0.84})_2Te_3$ with slightly different top and bottom surface (due to insertion of a 1 QL-thick $Sb_2Te_3$ buffer-layer between the film and InP substrate), which is believed to lead to non-degenerate LLs of top and bottom surface, thus asymmetric ($v = 0$ and 1) plateaux. The corresponding dips in $\sigma_{xx}$ are also marked. **c,** and **d,** Magnetic field dependence of $\sigma_{xy}$ for each sample, measured at 40 mK. Figures (a), (b), (c), and (d) are adapted from ref.[69]. **e,** Magnetic field dependence of $R_{xy}$ at different temperatures in magnetic field up to 34.5 T for a non-gated 8 QL-thick buffer-layer-based $Bi_2Se_3$ film with $MoO_3$/Se capping, which exhibits perfect quantization at $\frac{h}{e^2}$ at low temperatures. **f,** Magnetic field dependence of $R_{xx}$, which vanishes ($0.0 \pm 0.5$ $\Omega$) when Hall resistance quantizes to $h/e^2$. The signature of QH persists up to 70 K. (e) and (f) are adapted from ref. [57]. The corresponding angle dependence for **g,** $R_{xy}$ and **h,** $R_{xx}$ QHE. **i,** $R_{xx}$ and **j,** $R_{xy}$ at T = 0.35 K as a function of gate voltage $V_G$ at several magnetic field values from 0 to 44.5 T for a back-gated 10 QL-thick buffer-layer-based $Bi_2Se_3$ film with $MoO_3$/Se capping adapted from ref. [263]. Corresponding **k,** sheet ($\sigma_{xx}$) and **l,** Hall ($\sigma_{xy}$) conductance. $v = 1$ QHE is observed at $V_G \approx -15$ V to $-20$ V and non-saturating magnetoresistance with B is observed for $V_G \leq -21$ V as plotted in the inset of (i) for CNP and $V_G = -76$ V.

**Figure 4: QHE in MBE-grown ultra-low-Fermi-level Ti-doped Sb₂Te₃ TI films**

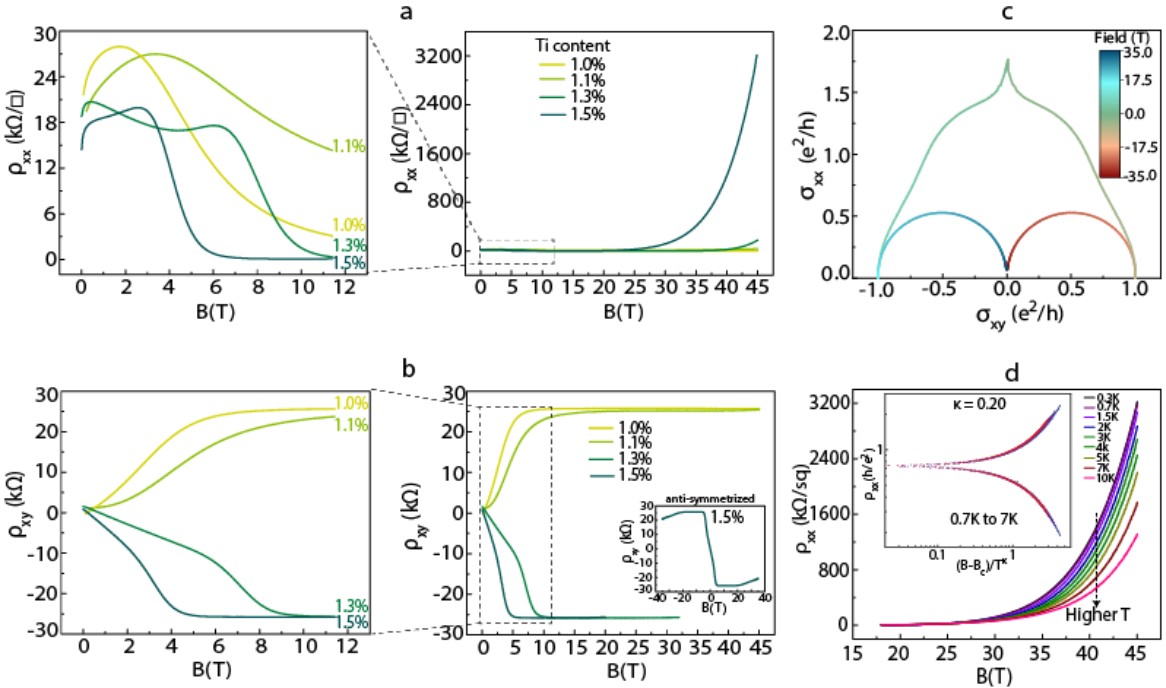

**a,** Magnetic field (0 to 45 T) dependence of $\rho_{xx}$ for 8 QL thick interface-engineered Sb₂Te₃ films with 1%, 1.1%, 1.3% and 1.5% Ti. The left panel shows $0 < B < 11$ T. All the films are grown on 20 QL-In₂Se₃/15 QL-(Sb₀.₆₅In₀.₃₅)₂Te₃ BL and capped with 15 QL SIT. **b,** Magnetic field (0 to 45 T) dependence of $\rho_{xy}$. The left panel shows $0 < B < 11$ T. The inset in b (right panel) shows the anti-symmetrized Hall resistivity of the 1.5% sample. **c,** The conductivity tensor flow plot ($\sigma_{xx}$ vs. $\sigma_{xy}$) for the 1.5% sample at $|B| < 35$ T. Each point on this plot corresponds to extracted ($\sigma_{xy}$, $\sigma_{xx}$) for each field. The field is incorporated as a color map in the plot. The points ($\pm e^2/h$,0) corresponds to the QH state, (0,0) corresponds to the (Hall) insulator phase, and ($\pm 0.5e^2/h$, $0.5e^2/h$) corresponds to the critical point of the transition between these two phases. The cusp around zero field indicates weak anti-localization. **d,** $\rho_{xx}$ as a function of magnetic field for the 1.5% sample at temperatures 300 mK through 10 K: all the curves pass through one another at the critical magnetic field ($B_c \approx 23.9$ T). The inset graph is the corresponding temperature scale-invariant plot, yielding $\kappa = 0.20 \pm 0.02$. These figures are adapted from ref.[60]

# Figure 5: QAHE in MBE-grown magnetic TI films

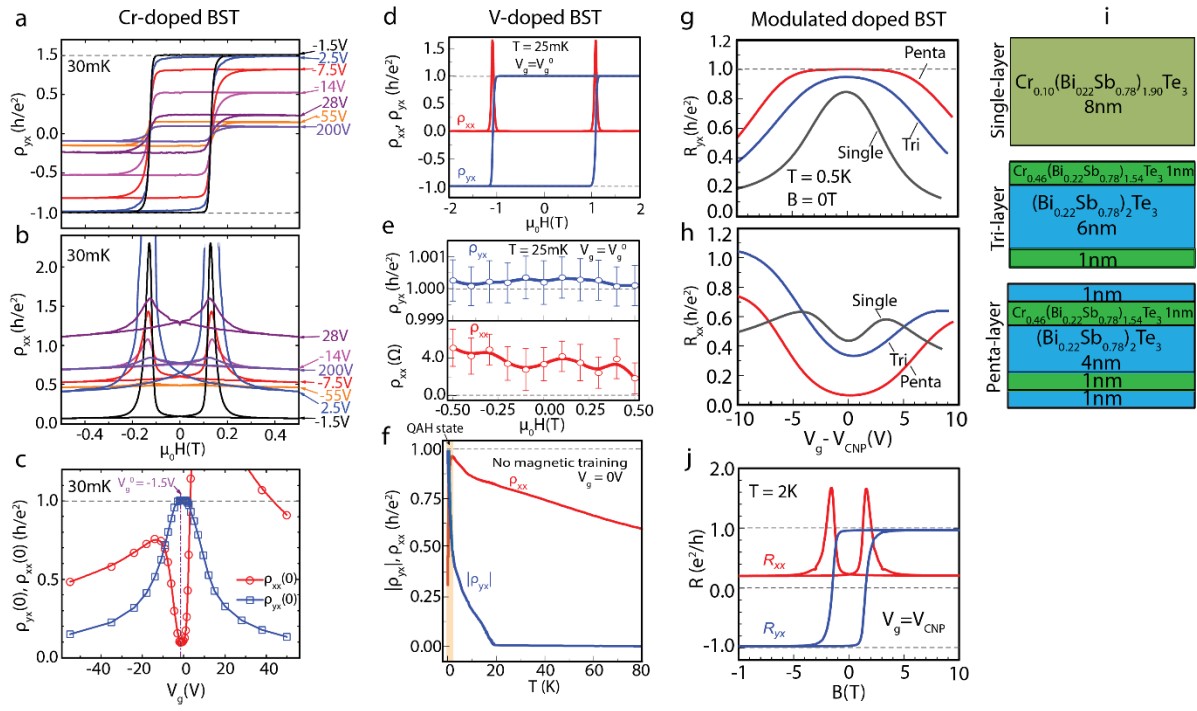

Magnetic field dependence of **a,** the Hall resistance $\rho_{yx}$ and **b,** the longitudinal sheet resistance ($\rho_{xx}$) for different gate voltages ($V_g$) and at 30 mK in a 5 QL-thick $Cr_{0.15}(Bi_{0.1}Sb_{0.9})_{1.85}Te_3$ film that exhibit QAHE. The shape and coercivity of the $\rho_{yx}$ hysteresis loop remain almost unchanged for different $V_g$. However, the height of loops changes significantly with $V_g$. $\rho_{yx}$ is nearly independent of the magnetic field, suggesting perfect ferromagnetic ordering. **c,** Gate voltage-dependence of $\rho_{yx}$ and $\rho_{xx}$ at zero field - labelled $\rho_{yx}(0)$ (blue squares) and $\rho_{xx}(0)$ (red circles), respectively. $\rho_{yx}$ shows a clear quantization ($\frac{h}{e^2}$) and $\rho_{xx}$ shows a sharp dip down to $0.098 \frac{h}{e^2}$ at the charge neutrality point ($V_g^0$), indicative of QAHE. Figures (a), (b), and (c) are adapted from ref. [61] **d,** Magnetic field dependence of $\rho_{xx}$ and $\rho_{yx}$ at the charge neutral point ($V_g^0$) in a 4 QL-thick $(Bi_{0.29}Sb_{0.71})_{1.89}V_{0.11}Te_3$ film that exhibits an enhanced QAHE compared to the Cr-doped film. **e,** Expanded view of the magnetic field-dependence of $\rho_{yx}$ (top panel) and $\rho_{xx}$ (bottom panel) at $|\mu_0 H| < 0.5$ T for the same film. At zero magnetic field, $\rho_{yx} = 1.00019 \pm 0.00069 \frac{h}{e^2}$ and $\rho_{xx}$ vanishes to $0.00013 \pm 0.00007 \frac{h}{e^2}$. **f,** $\rho_{xx}$ (red curve) and $|\rho_{yx}|$ (blue curve) as a function of temperature without magnetic training. Figures (d), (e), and (f) are adapted from ref.[62]. $V_g$ dependence of **g,** $\rho_{yx}$ and **h,** $\rho_{xx}$ of modulation magnetic-doped $Cr_x(Bi_{1-y}Sb_y)_{2-x}Te_3$ structures: single-layer (gray line; schematic shown in **i,** first panel), tri-layer (blue line; schematic shown in **i,** second panel), and the penta-layer (red line; schematic shown in **i,** third panel) at 0.5 K in the absence of external magnetic field. **j,** Magnetic field dependence of $\rho_{yx}$ and $\rho_{xx}$ for penta-layer (for the most optimized sample with $x = 0.57$, $y = 0.74$) for $V_g^0$ at 2 K. Quantized Hall resistance is observed up to 1 K, where the residual $\rho_{xx}$ is $0.081 \frac{h}{e^2}$, which is slightly higher than $0.017 \frac{h}{e^2}$ at 0.5 K. At 2 K, $\rho_{yx}$ shows a reasonable quantization, $\pm 0.97 \frac{h}{e^2}$. Then, at 4.2 K, it deviates from perfect quantization ($\rho_{yx} \approx \pm 0.87 \frac{h}{e^2}$) with a Hall angle of 66.1°. 2 K is so far the highest temperature at which (albeit, approximate) QAH is observed. Figures (g), (h), (i), and (j) are adapted from ref.[63]

# Figure 6: AH/QAH loops of various magnetic TI films

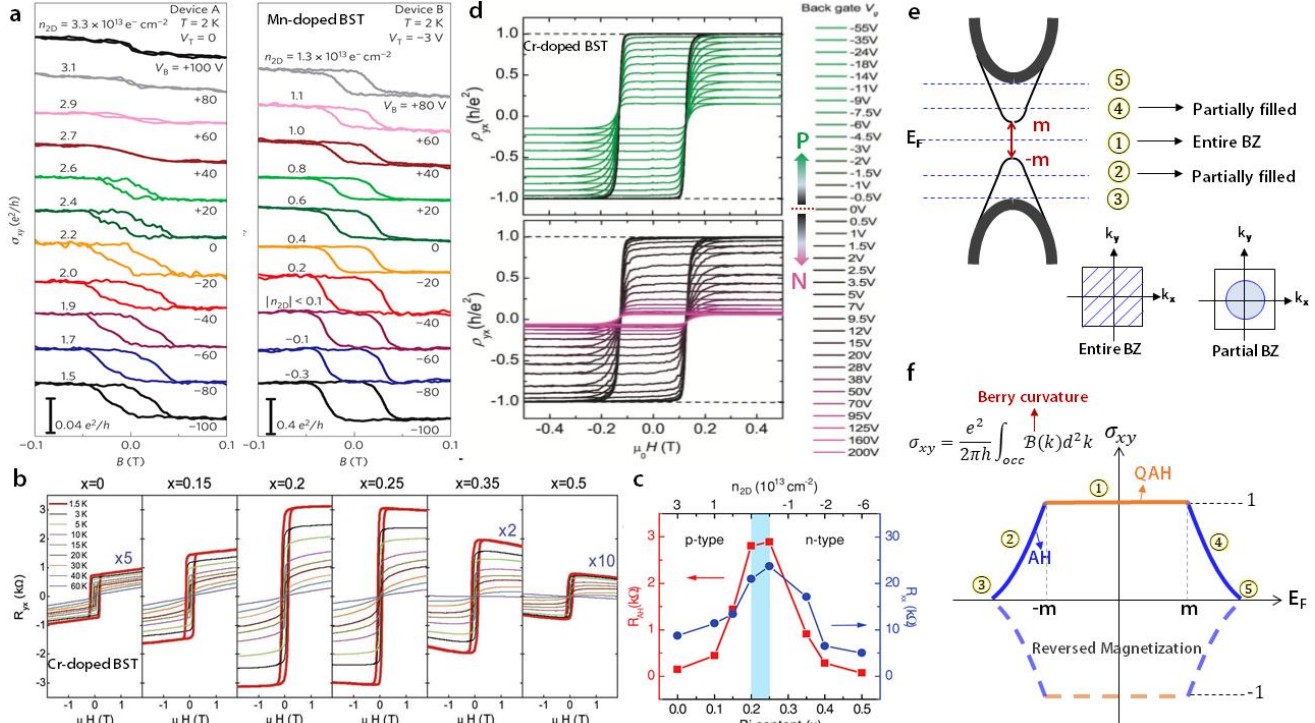

**a,** Sign of the AH hysterisis loops remains the same for the entire carrier range from n to p-type for MBE-grown Mn-doped $(Bi,Sb)_2Te_3$ films. The disappearance of the loop at high carrier density, when the Fermi level is deep in the bulk conduction band, indicates that the ferromagnetism is not mediated by bulk carriers (adapted from ref. [267]). **b,** Sign of the AH hysterisis loops remains the same for the entire Bi doping concetration in Cr-doped $(Bi_xSb_{1-x})_2Te_3$ and is independent of the carrier type, which can be tuned from p to n upon adding Bi, as shown in **c** (adapted from ref. [266]). **d,** The positive QAH loops for Cr-doped $(Bi,Sb)_2Te_3$ films (adapted from ref.[61]). The sign of the loop remains the same for the entire gate-voltage range and even when the system is far from the quantized regime. **e,** Schematic band structure of a magnetically-doped TI. Magnetic doping/proximitizing induces an exchange gap of 2m as well as a Berry curvature in the band structure. The Hall conductance can be written as $\sigma_{xy} = \frac{e^2}{2\pi h}\int_{occ}\mathcal{B}d^2k$ (i.e. the integral of Berry curvature $\mathcal{B}$ over the entire occupied states). If the integration covers the entire Brillouin-zone, which is the case for QAHE where $E_F$ is located in the exchange gap of an MTI, the Hall conductance becomes an integer. Otherwise, it takes any number from 0 to 1(like in AH case). Whether the quantization value is $\frac{e^2}{h}$ or $-\frac{e^2}{h}$ depends on the sign of the exchange gap and how magnetization is induced (shown in **f**). The numbers in (e) mark different locations of $E_F$ and the corresponding AH response in **f**. For example, location 1 is when the $E_F$ is in the exchange gap and the integration is over the entire Brillouin zone (BZ) (hashed BZ), whereas location 2 is when the band is partially filled, and the integration covers only a portion of BZ (shown by a circle in the BZ).

## Figure 7: Effect of Ca-doping on interface-engineered Bi₂Se₃ films

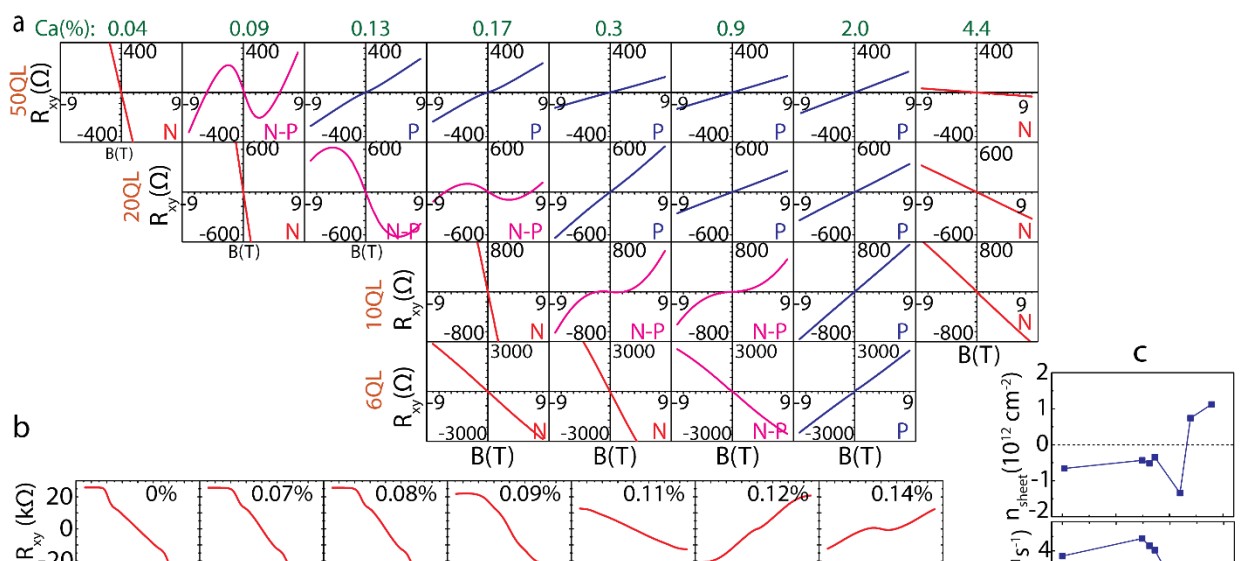

**a,** Magnetic field-dependence of $R_{xy}$ of Ca-doped Bi₂Se₃ (with Se cap) films for various Ca concentrations and different thicknesses. n-type (negative slope), p-type (positive slope), and non-linear n-p mixed curves are colored as red, blue, and pink, respectively. As Ca concentration increases, all the films transition from n- to p-type through an n-p mixed regime, and then eventually become n-type again except for the 6 QL film which becomes insulating, instead. **b,** Magnetic field-dependence of $R_{xy}$ and $R_{xx}$ and the evolution of QH in 8 QL-thick buffer-layer-based Bi₂Se₃ films (capped by MoO₃ and Se) with different Ca doping at 300 mK. **c,** Extracted 2D sheet carrier density ($n_{sheet}$) in top panel and mobility ($\mu$) in bottom panel for different Ca doping levels. $\mu$ sharply decreases when the carrier type changes from n- to p-type. All the figures are adapted from ref. [59]

**Figure 8: Quantized Faraday and Kerr rotations in interface-engineered Bi₂Se₃ films**

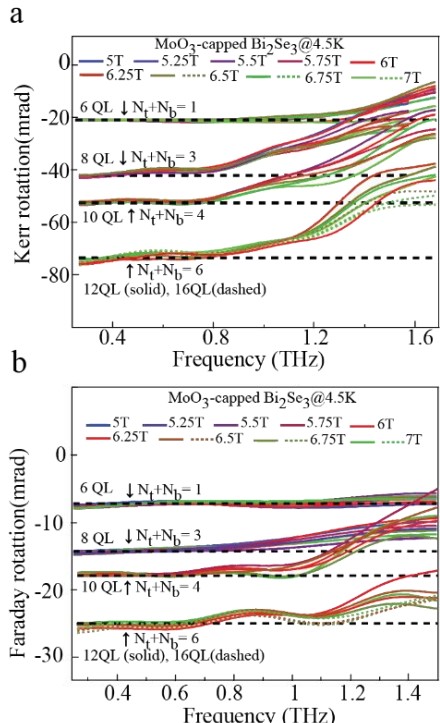

**a,** Quantized Kerr rotation and **b,** Quantized Faraday rotation in buffer-layer-based Bi₂Se₃ films with MoO₃ and Se capping layers for different thicknesses. Dashed black lines are theoretical expectation values corresponding to the indicated filling factors of the surface states. Adapted from ref. [76]

**Figure 9: Topological and metal insulator transitions in interface-engineered (Bi₁₋ₓInₓ)₂Se₃ films**

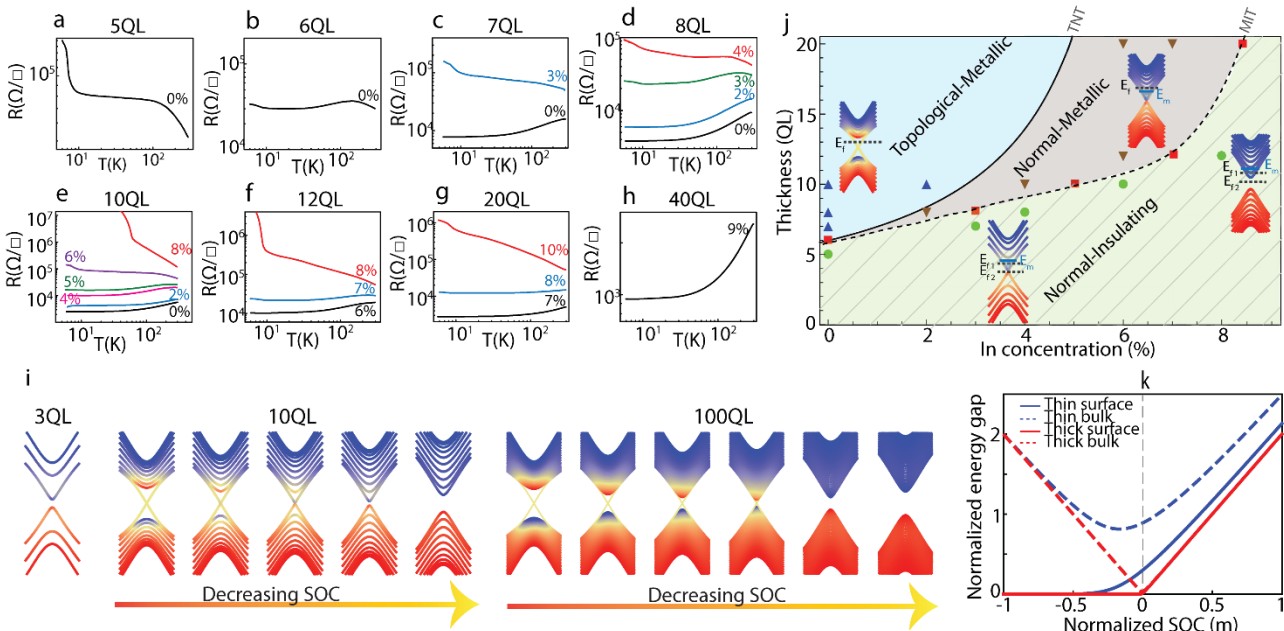

Temperature-dependence of sheet resistance for **a,** 5 QL with 0% In, **b,** 6 QL with 0% In, **c,** 7 QL with 0% and 3% In, **d,** 8 QL with 0%, 2%, 3% and 4% In, **e,** 10 QL with 0%, 2%, 4%, 5%, 6% and 8% In, **f,** 12 QL with 6%, 7% and 8% In, **g,** 20 QL with 7%, 8% and 10% In, and **h,** 40 QL with 9% In. **i,** Schematic of the TPT (topological phase transition) process for different thicknesses and as a function of SOC weakening (due to In substitution). **j,** Phase diagram for the interface-engineered samples: the dashed line is a guide to eye for the MIT (metal insulator transition) boundary. Blue triangles correspond to topological-metallic data points where $k_F l > 1$, green circles are normal-insulating data points with $k_F l < 1$, brown triangles show normal-metallic data points with $k_F l > 1$, red squares with $k_F l = 1$ and the MIT line goes through these points. The Ioffe-Regel criterion can be used to quantitatively identify the MIT point. Following this criterion, a material becomes metal if $k_F l > 1$ and insulator if $k_F l \leq 1$, where 2D Fermi vector $k_F = (2\pi n_{sheet})^{1/2}$ and mean-free path $l = (\hbar\mu/e)(2\pi n_{sheet})^{1/2}$ [275,276]. In the normal insulating region, $E_{F1}$ and $E_{F2}$ represent two possible locations of the surface Fermi level, the former below the mobility edge (represented by $E_m$) and the latter inside the surface hybridization gap, which is an ideal case, where the MIT and TNT lines will overlap. On the other hand, without the interface engineering scheme, the surface Fermi level is high and above the bottom of the conduction band, and the film remains metallic even after the TNT line and the MIT line cannot be detected via transport measurement [14,163]. **k,** Simulated phase diagram. The critical point for TNT is well-defined in thick samples, but it is not indisputably definable in thin samples. The vertical dashed line (m = 0) shows the phase boundary in the infinite-size limit. Both axes are normalized by the Dirac velocity in this model (adapted from ref.[78]).

# Acknowledgements

We would like to thank Hassan Shapourian for his insightful discussions. This work is funded by National Science Foundation (NSF) Grant No. DMR2004125, Army Research Office (ARO) Grant No. W911NF2010108, MURI Grant No. W911NF2020166 and the center for Quantum Materials Synthesis (cQMS), funded by the Gordon and Betty Moore Foundation's EPiQS initiative through Grant No. GBMF10104.

# Authors information

## Affiliations

Department of Materials Science and Engineering, Rutgers, The State University of New Jersey, Piscataway, New Jersey 08854, United States

Maryam Salehi

Center for Quantum Materials Synthesis and Department of Physics and Astronomy, Rutgers, The State University of New Jersey, Piscataway, New Jersey 08854, United States

Xiong Yao and Seongshik Oh

## Competing interests

The authors declare no competing financial interest.

## Corresponding author

Correspondence to Seongshik Oh. *E-mail: ohsean@physics.rutgers.edu