# Peer review of "From classical to quantum regime of topological surface states via defect engineering"

_SciPost Physics Lecture Notes_

## Round 1 · Referee Report · Anonymous (Referee 1) · 2022-3-1

Strengths

Very important topic written by the leading group in this topic

Comprehensive review concentrating on almost all aspects of thin film preparation of different kinds of topological materials

Presentation of fundamental problems and their resolution

Weaknesses

Discussion on how band banding effects are relevant is confusing. The discussion on p. 12, Sec. III is confusing. Its not clear precisely the considerations that go into determine the flat band condition for instance. For instance, the logic leading to the conclusions in the sentence "If we calculate 𝑛𝑠𝑠 for Bi2Se3 with its conduction band minimum at almost 200 meV above the Dirac..." is not clear. (Top . 13) How is it this density gives the flat band condition.

There is no discussion of the preparation of thin films of Weyl semimetals. There were notable theoretical predictions, but no efforts that I am aware of. Some discussion on these possibilities would be appropriate in the overview about future directions.

Report

Excellent review. As far as I know no other reports like it. Should be published after small changes.

Requested changes

See above.

  • validity: top
  • significance: top
  • originality: high
  • clarity: good
  • formatting: excellent
  • grammar: reasonable

Author:  Xiong Yao  on 2022-04-14  [id 2384]

(in reply to Report 1 on 2022-03-01)
Category:
answer to question

We would like to thank the referee for emphasizing the significance of our review paper, and for recommending publication after a minor revision.
We also would like to thank the referee for mentioning the discussion on band bending effects.
The flat band condition can be understood from the attached Figure 1. Given that the Mott criterion fixes $E_F$ to be at the conduction band minimum (confirmed by previous report) and knowing the surface state carrier density, together these fix the band bending as follows. First, assume that at the surface $E_F$ barely touches the CB minimum. Then assuming a Fermi velocity (4.0 *$10^5$ m/s) and the energy of the CB minimum relative to the Dirac point (210 meV), the surface carrier density can be calculated for one surface to be $n_{SS}$=5.0*$10^{12}$/$cm^2$ by the formula given on Page 12 and 13. Since EF at the surface and in the bulk is at the CB minimum, the bands must be flat, and band bending is negligible (see attached Figure 1). Therefore, $n_{SS}$=5.0*$10^{12}$/$cm^2$ gives the flat band condition. Either a higher or lower carrier density would give rise to an accumulation or depletion of carriers and band bending near the surface.
Due to limitations of space and scope, we are not able to extend all the details about band bending effects in topological insulators in this paper. These details about the band bending effects are included in the cited reference 84 "Matthew Brahlek et.al, Solid State Communications, 215–216, 54-62 (2015)", which is a review paper previously published by our group.
We agree with the referee that thin film growth of Weyl semimetals is important to the quantum materials community. However, the topic of the current review paper is topological insulators rather than semimetals. Discussion on Weyl semimetals are beyond the scope of this paper and should be included in another separate review paper.

Attachment:

---

## Editorial Decision

resubmitted